# NEW INSIGHTS FOR THE STABILITY-PLASTICITY DILEMMA IN ONLINE CONTINUAL LEARNING

**Dahuin Jung[1], Dongjin Lee[1], Sunwon Hong[2], Hyemi Jang[1], Ho Bae[3,*], Sungroh Yoon[1,4,*]**

[1]Department of Electrical and Computer Engineering, Seoul National University,
[2]DeepBrain AI Inc., [3]Department of Cyber Security, Ewha Womans University, and
[4]Interdisciplinary Program in Artificial Intelligence, Seoul National University, Seoul 08826, Korea

annajung0625@snu.ac.kr  ldj9510@snu.ac.kr  harry@deepbrainai.io
wkdal9512@snu.ac.kr  hobae@ewha.ac.kr  sryoon@snu.ac.kr

## ABSTRACT

The aim of continual learning is to learn new tasks continuously (*i.e.*, plasticity) without forgetting previously learned knowledge from old tasks (*i.e.*, stability). In the scenario of online continual learning, wherein data comes strictly in a streaming manner, the plasticity of online continual learning is more vulnerable than offline continual learning because the training signal that can be obtained from a single data point is limited. To overcome the stability-plasticity dilemma in online continual learning, we propose an online continual learning framework named multi-scale feature adaptation network (MuFAN) that utilizes a richer context encoding extracted from different levels of a pre-trained network. Additionally, we introduce a novel structure-wise distillation loss and replace the commonly used batch normalization layer with a newly proposed stability-plasticity normalization module to train MuFAN that simultaneously maintains high plasticity and stability. Mu-FAN outperforms other state-of-the-art continual learning methods on the SVHN, CIFAR100, miniImageNet, and CORe50 datasets. Extensive experiments and ablation studies validate the significance and scalability of each proposed component: 1) multi-scale feature maps from a pre-trained encoder, 2) the structure-wise distillation loss, and 3) the stability-plasticity normalization module in MuFAN. Code is publicly available at https://github.com/whitesnowdrop/MuFAN.

## 1 INTRODUCTION

Humans excel in learning new skills without forgetting what they have previously learned over their lifetimes. Meanwhile, in continual learning (CL) (Chen & Liu, 2018), wherein a stream of tasks is observed, a deep learning model forgets prior knowledge when learning a new task if samples from old tasks are unavailable. This problem is known as *catastrophic forgetting* (McCloskey & Cohen, 1989). In recent years, promising research has been conducted to address this problem (Parisi et al., 2019). However, excessive retention of old knowledge impedes the balance between preventing forgetting (*i.e.*, stability) and acquiring new concepts (*i.e.*, plasticity), which is referred to as the stability-plasticity dilemma (Abraham & Robins, 2005). In this study, we cover the difference in the stability-plasticity dilemma encountered by online CL and offline CL and propose a novel approach that addresses the stability-plasticity dilemma in online CL.

Most offline CL methods aim at less constraining plasticity in the process of preventing forgetting instead of improving it because obtaining high plasticity through iterative training is relatively easy. However, as shown in Figure 1, the learning accuracy (showing plasticity) of online CL is way lower than that of offline CL, with a gap of 10–20% on all three CL benchmarks. That is, for online CL, wherein data comes in a streaming manner (single epoch), an approach that aims at suppressing excessive forgetting in the process of enhancing plasticity is required. For it, we propose a multi-scale feature adaptation network (MuFAN), which consists of three components to obtain high stability and plasticity simultaneously: 1) multi-scale feature maps exploited from shallow to deeper layers of a pre-trained model, 2) a novel structure-wise distillation loss across tasks, and 3) a

---

*Corresponding Authors

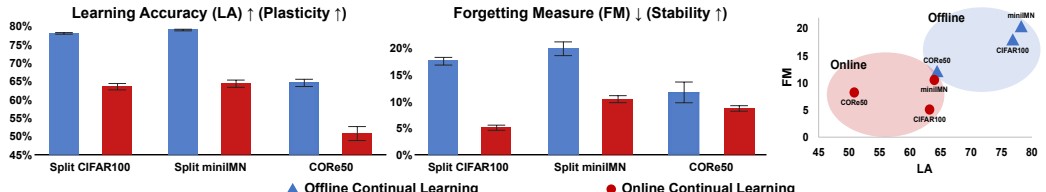

Figure 1: Comparison results of ER-Ring on three CL benchmarks in offline and online CL in bar (left and middle) and scatter (right) plots. For offline CL, plasticity is relatively high, whereas stability is low. In contrast, for online CL, stability is relatively high, whereas plasticity is low. It shows the difference in trend between offline and online CL in terms of the stability-plasticity dilemma. Further analysis of the difference in the trend is provided in Appendix A.

novel stability-plasticity normalization module considering both the retention of old knowledge and fast adaptation in a parallel way.

First, using pre-trained representations has become common in computer vision (Patashnik et al., 2021; Kolesnikov et al., 2020; Ranftl et al., 2020) and natural language processing (Radford et al.; Peters et al., 2018). Meanwhile, the use of pre-trained representations in CL is still naïve, for example, using an ImageNet-pretrained ResNet as a backbone (Yin et al., 2021; Wu et al., 2021; Park et al., 2021; Hayes et al., 2020), which limits the structure or size of the pre-trained model that can be used. As shown in Figure 2 (a), instead of using a pre-trained model as a backbone, we propose an approach that uses the pre-trained model as an encoder to obtain a richer multi-context feature map. Rather than leveraging a raw RGB image, we accelerate classifier training by leveraging an aggregated feature map from the meaningful spaces of the pre-trained encoder. We also verify the scalability of the aggregated multi-scale feature map by integrating it into existing online CL methods.

Second, we present a novel structure-wise distillation loss to suppress catastrophic forgetting. Most distillation losses in CL are point-wise (Chaudhry et al., 2019; Buzzega et al., 2020), and point-wise distillation is indeed effective in alleviating forgetting. Another effective way to preserve knowledge in a classification task is through a relationship between tasks, especially in a highly non-stationary online continual setting. As described in Figure 2 (b), we propose a novel structure-wise distillation loss that can generate an extra training signal to alleviate forgetting using the relationship between tasks in a given replay buffer.

Finally, the role of normalization in CL has been investigated in recent studies (Pham et al., 2021b; Cha et al., 2022; Zhou et al., 2022). In the field of online CL, switchable normalization (SN) (Luo et al., 2018) and continual normalization (CN) (Pham et al., 2021b), which use both minibatch and spatial dimensions to calculate running statistics, have led to improvement in the final performance. However, we observed that these approaches do not fully benefit from either batch normalization (BN) (Ioffe & Szegedy, 2015) or spatial normalization layers. To address this problem, we propose a new stability-plasticity normalization (SPN) module that sets one normalization operation efficient for plasticity and another normalization operation efficient for stability in a parallel manner.

Through comprehensive experiments, we validated the superiority of MuFAN over other state-of-the-art CL methods on the SVHN (Netzer et al., 2011), CIFAR100 (Krizhevsky et al., 2009), mini-ImageNet (Vinyals et al., 2016), and CORe50 (Lomonaco & Maltoni, 2017) datasets. On CORe50, MuFAN significantly outperformed the other state-of-the-art methods. Furthermore, we conducted diverse ablation studies to demonstrate the significance and scalability of the three components.

## 2 RELATED WORK

### 2.1 CONTINUAL LEARNING

To date, in the field of CL, the utility of a pre-trained model is relatively limited to obtaining the last feature map projected in a meaningful space by using a pre-trained model as a feature extractor. CL methods that use a pre-trained model can be categorized into two groups based on the scenario.

The first utilizes an ImageNet-pretrained ResNet as a backbone for CL on fine-grained or video-based datasets to increase the base accuracy (Yin et al., 2021; Wu et al., 2021; Park et al., 2021; Hayes et al., 2020). In general, these methods update an entire classifier continuously during training on

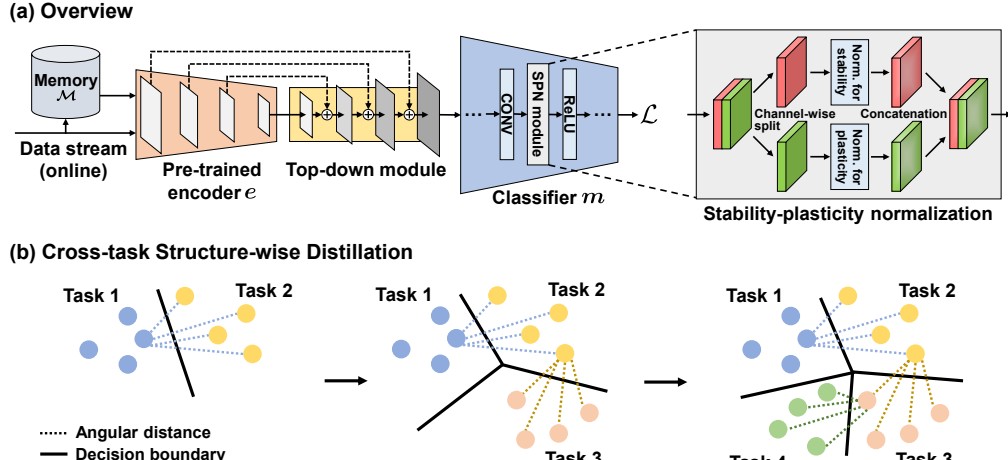

Figure 2: (a) Overview of MuFAN. MuFAN consists of a pre-trained (fixed) encoder $e$ and a classifier $m$. MuFAN obtains an aggregated multi-scale feature map using a top-down module and replaces every BN in $m$ with the proposed SPN. (b) Simplified visualization of how the proposed structure-wise potential function builds a structural relation between tasks during training on a stream of tasks.

a stream of tasks rather than freezing a feature extractor. The methods in the second group mostly freeze an ImageNet-pretrained feature extractor after fine-tuning in a base training session (Hayes & Kanan, 2020; Maracani et al., 2021; Ganea et al., 2021; Zhang et al., 2021; Cheraghian et al., 2021; Wu et al., 2022). They focused on finding a novel CL method that can efficiently train a classification layer. Deep SLDA (Hayes & Kanan, 2020) trained a classification layer using linear discriminant analysis, and CEC (Zhang et al., 2021) trained a classification layer to transfer context information for adaptation with a graph model. Lately, a continual semantic segmentation method, called RECALL (Maracani et al., 2021), which uses a pre-trained model as an encoder was proposed. However, MuFAN is the first online CL method that utilizes a richer multi-scale feature map from different layers of the pre-trained encoder to obtain a strong training signal from a single data point.

## 2.2 NORMALIZATION LAYER

BN (Ioffe & Szegedy, 2015) has been used as a standard to improve and stabilize the training of neural networks by normalizing intermediate activations. To perform normalization, numerous alternatives to BN, which effectively aid the learning of neural networks in various tasks, have been proposed. A normalization layer can be categorized into three approaches based on the method of calculating the running mean and variance along convolutional features for normalization: 1) BN and its variants, batch renormalization (BRN) (Ioffe, 2017) and representative batch normalization (RBN) (Gao et al., 2021), calculate the running statistics along the minibatch dimension, 2) instance normalization (IN) (Ulyanov et al., 2016), group normalization (GN) (Wu & He, 2018), and layer normalization (LN) (Ba et al., 2016) calculate the running statistics along the spatial (channel or layer) dimension, and 3) SN (Luo et al., 2018) and CN (Pham et al., 2021b) are hybrid methods that utilize both the running statistics along the minibatch and spatial dimensions in either a mixture or sequential manner. A detailed explanation of each normalization layer can be found in Appendix B. Unlike SN and CN, our proposed SPN splits a convolutional feature map along the channel dimension and applies a different normalization operation to each feature map in a parallel manner. Recently, several studies have suggested using multiple normalization layers in a parallel manner (Pan et al., 2018; Xie et al., 2020; Wang et al., 2021; Zada et al., 2022). However, mostly, they utilize multiple normalization layers in a way of selecting one from them according to a purpose. The multiple normalization module has been explored to improve the generalization of the model in the field of adversarial training (Xie et al., 2020) and insufficient data (Zada et al., 2022).

## 3 METHOD

In this section, we first explain why MuFAN makes use of a pre-trained network as an encoder and an aggregated multi-scale feature map rather than a single feature map for online CL. We then introduce

a novel structure-wise distillation loss that regularizes the shift in structural relations between tasks. Finally, we propose the SPN module that can be easily integrated to CL models to maintain high stability and plasticity simultaneously.

**Notation.** MuFAN consists of two networks: a pre-trained encoder $e$ and a classifier $m$, and an experience replay buffer $\mathcal{M}$, as illustrated in Figure 2. The classifier $m$ consists of a feature extractor and a classification layer. Suppose that there are $T$ different tasks with respect to data streams $\{\mathcal{D}_1, \cdots \mathcal{D}_T\}$, where $\mathcal{D}_i = \{x_i, y_i\}$ for $i \in \{1, \cdots, T\}$ (labeled sample), We denote a tuple of $N$ data samples from $j$th task by $\mathcal{X}_j^N = (x_j^1, x_j^2, ..., x_j^N)$. For SPN, the feature map of a minibatch from $m$ is denoted as $\boldsymbol{a} \in \mathbb{R}^{B \times C \times W \times H}$, where $B$ is the minibatch size, $C$ is the number of channels, $W$ is the width, and $H$ is the height.

## 3.1 Multi-scale Features from a Pre-trained Encoder

Recent studies in computer vision and natural language processing have shown that utilizing pre-trained models in various ways improves the model's capacity and expands a range of available tasks (Patashnik et al., 2021; Kolesnikov et al., 2020; Ranftl et al., 2020; Peters et al., 2018; Radford et al.). However, in CL, the utility of a pre-trained model is still limited to using a pre-trained network as a backbone. For CL, there is no such approach that utilizes a pre-trained network to obtain an aggregated multi-scale feature map constructed by augmenting contextual information extracted from different levels.

To consider why an aggregated multi-scale feature map from a pre-trained model improves classification, we first need to know the characteristics of the feature map from the pre-trained encoder, excluding multi-scale. For a raw image, a pixel value of 127 is not semantically close to a pixel value of 126 or 128. When leveraging a raw RGB image, a classifier in online CL should learn not only discriminative features but also semantic embedding within single epoch training. However, by leveraging the feature map from the semantic space of the pre-trained encoder, we can accelerate classifier training. Then, we should consider why we need to utilize multi-scale feature maps rather than a single feature map such as a first or last feature map. Multi-scale feature maps have been demonstrated to be beneficial in object segmentation (Lin et al., 2019; Guo et al., 2020; He et al., 2019) and also recently in generative models (Sauer et al., 2021; 2022). The low-level features extracted by shallow layers encode more pattern-wise and general information, whereas the high-level features extracted by deeper layers contain more contextual information that focuses on prominent features. That is, the aggregated multi-scale feature map from shallow to deeper layers of the pre-trained encoder can compensate for the shortcomings of each layer. In an online setting, the classifier needs to learn as much information as possible within the restricted training iteration. For that, MuFAN takes advantage of the multi-scale feature map that provides condensed information obtained by diverse layers.

The concept of augmenting the context across different levels of a deep neural network consists of two approaches: bottom-up and top-down. As illustrated in Figure 2 (a) and Figure 3 in Appendix C, we obtain convolutional feature maps from four layers of the pre-trained model at different resolutions. The bottom-up module propagates more pattern-wise and general features into deeper layers using max-pooling, whereas the top-down module propagates more semantic features to shallow layers using bilinear upsampling. In this process, the bottom-up module propagates only specific features and discards the remaining ones. Between these two modules, MuFAN uses the top-down module to fully exploit the benefits of multi-scale hierarchical features. By taking the aggregated multi-scale feature map using the top-down module as input, the classifier $m$ can carefully integrate the information required for classification among the rich features. We tested the possibility of a wide range of pre-trained models as an encoder, such as a large self-supervised model or an object detection model. For mixing strategies between multi-scale feature maps, we follow differentiable random projections, yielding a U-Net-like architecture, proposed by Sauer et al. (2021), as described in more detail in Appendix C.

## 3.2 Cross-task Structure-wise Distillation Loss

Most distillation losses in CL have been point-wise; for each data point, they regularized the change by distilling individual outputs. More precisely, the conventional point-wise distillation losses for CL essentially reduce forgetting by using an original prediction output as a soft label.

The conventional distillation loss based on the replay buffer $\mathcal{M}$ (Pham et al., 2020) is expressed as follows:

$$\mathcal{L}_{\text{D-CTN}} = \sum_{i=1} l_{\text{D-CTN}}(m_j^*(x_j^i), m_t(x_j^i)) \in \mathcal{M}, \tag{1}$$

where $l_{\text{D-CTN}}$ is a Kullback-Leibler divergence, $t$ represents a current task, $j$ represents a task ID, and $m_j^*$ represents the final classifier at the end of task $j$. The point-wise loss is commonly used in many CL methods to alleviate forgetting (Buzzega et al., 2020; Pham et al., 2020; 2021a).

Based on knowledge distillation (Chen et al., 2018; Park et al., 2019; Fang et al., 2020; Xu et al., 2021), in addition to the point-wise information, we propose another regularization loss that can create an extra training signal irrelevant to label information. Unlike conventional approaches, we present a structure-wise distillation loss for CL. We compute a structure-wise potential $\psi$ for a tuple of data samples across tasks from $\mathcal{M}$ and distill relations through the potential. Our distillation loss $\mathcal{L}_{\text{D-CSD}}$ regularizes forgetting by forcing an angular distance to be maintained between tasks. The distillation loss is defined as follows:

$$\mathcal{L}_{\text{D-CSD}} = \sum_{j=2}^{t-1} \sum_{i=1}^{N} l(\psi_{m_{t-1}^*}(z_{j-1}^i, \mathcal{Z}_j^N), \psi_{m_t}(z_{j-1}^i, \mathcal{Z}_j^N)), \tag{2}$$

where $z = e(x)$ (and $\mathcal{Z} = e(\mathcal{X})$) is the multi-scale feature map from $e$ with the top-down module, $l$ is a cross-entropy loss that penalizes the difference between the previous and current models, $m_{t-1}^*$ and $m_t$. At the end of each task, $N$ samples per task are randomly selected from $\mathcal{M}$. $\psi$ represents a structure-wise potential function that measures the relational energy of the cross-task tuple.

The distillation loss $\mathcal{L}_{\text{D-CSD}}$ can force the current model $m_t$ to maintain the same relationship between cross-task samples. We suggest a potential function that measures the angular distance between samples of a cross-task. Our structure-wise potential function $\psi$ returns an $N$-dimensional vector as

$$\psi_m(z_{j-1}^i, \mathcal{Z}_j^N)) = \sigma(<m(z_{j-1}^i), m(z_j^1)>, \cdots, <m(z_{j-1}^i), m(z_j^N)>), \tag{3}$$

where $\sigma$ represents a softmax function and $<,>$ represents cosine similarity (CS). As illustrated in Figure 2 (b), our loss function seeks to maintain the angular distance between the samples of consecutive tasks. In this process, we penalize the angular differences between tasks $j - 1$ and $j$, and tasks $j$ and $j + 1$, where task $j$ is then forced to maintain the same relationship with both $j - 1$ and $j + 1$. We empirically confirmed that maintaining the angular distance with the previous and next tasks, $j - 1$ and $j + 1$, respectively, is more advantageous in terms of stability than that with a single task. Our structure-wise distillation loss can be used with conventional point-wise distillation losses to further improve performance. The overall objective of the proposed method is expressed as follows:

$$\mathcal{L} = \mathcal{L}_{\text{CE}} + \mathcal{L}_{\text{ER}} + \lambda_{\text{D-CTN}}\mathcal{L}_{\text{D-CTN}} + \lambda_{\text{D-CSD}}\mathcal{L}_{\text{D-CSD}}, \tag{4}$$

where $\mathcal{L}_{\text{CE}} = \sum_{i=1} l(y^i, m_t(h^i)) \in \mathcal{D}_t$ is a cross-entropy loss for the current task $t$ and $\mathcal{L}_{\text{ER}} = \sum_{i=1} l_{\text{ER}}(y^i, m_t(x^i)) \in \mathcal{M}$ is a cross-entropy loss for the past tasks. Also, the inputs of $\mathcal{L}_{\text{ER}}$ and $\mathcal{L}_{\text{D-CTN}}$ are replaced with $h$ in our objective. $\lambda_{\text{D-CTN}}$ and $\lambda_{\text{D-CSD}}$ are balancing factors for each loss. By maintaining the angular similarity of task embedding spaces between consecutive tasks with $\mathcal{L}_{\text{D-CSD}}$, forgetting can be effectively reduced. While several studies (Hou et al., 2019; Wu et al., 2021) apply an adaptive weight for effective distillation as the task increases, our $\mathcal{L}_{\text{D-CSD}}$ naturally grows as the task increases without additional adaptation, rendering the use of an adaptive weight unnecessary.

### 3.3 STABILITY-PLASTICITY NORMALIZATION MODULE

We propose a novel online CL normalization module that splits the feature map $\boldsymbol{a}$ of the classifier $m$ into halves along the channel dimension and applies a different normalization operation to each half feature map. A few recent works have investigated the role of normalization in CL and proposed a new normalization module for CL (Pham et al., 2021b; Cha et al., 2022; Zhou et al., 2022). However, to the best of our knowledge, this is the first study that sets normalization layers with different strengths in a parallel way to obtain high stability and plasticity simultaneously. By setting one normalization operation efficient for stability and another normalization operation efficient for plasticity in a parallel manner, we observed a significant improvement in the final performance.

Table 1: Comparison results on four CL benchmarks. The same backbone and 50 memory slots per task are used by all methods (MF: the aggregated multi-scale feature map from the pre-trained encoder as input). Bold fonts represent the best performance in each evaluation metric.

| Method | Split SVHN | | | Split CIFAR100 | | |
|--------|------------|------------|------------|------------|------------|------------|
| | ACC ↑ | FM ↓ | LA ↑ | ACC ↑ | FM ↓ | LA ↑ |
| GEM | 82.30±3.86 | 12.16±4.81 | 91.06±1.46 | 57.89±0.98 | 8.62±0.28 | 63.01±1.30 |
| ER-Ring | 91.68±1.17 | 5.26±1.10 | 95.48±0.68 | 61.32±0.86 | 5.16±0.50 | 63.20±0.84 |
| MIR | 91.22±0.43 | 6.18±0.54 | 96.16±0.14 | 64.97±0.94 | 7.78±1.47 | 70.03±1.90 |
| CTN | 92.14±1.84 | 3.08±1.34 | 94.42±2.69 | 67.04±2.86 | 4.25±3.00 | 69.21±0.48 |
| DualNet | 93.88±0.51 | 3.04±0.43 | 96.18±0.98 | 72.61±0.76 | **3.82±0.63** | 74.65±0.40 |
| ER-Ring w/ MF | 92.30±0.31 | 5.76±1.39 | 96.86±2.06 | 69.33±1.61 | 8.78±1.26 | 77.41±0.65 |
| CTN w/ MF | 93.53±1.22 | 3.90±1.11 | 95.97±1.37 | 72.26±0.87 | 5.30±0.68 | 76.27±0.64 |
| DualNet w/ MF | 94.06±0.55 | 3.48±1.01 | 96.58±2.02 | 74.66±0.58 | 5.01±0.58 | 76.71±0.29 |
| MuFAN | **94.76±0.68** | **2.90±0.60** | **97.10±0.36** | **75.86±0.35** | 4.24±0.26 | **78.58±0.41** |

| Method | Split miniIMN | | | CORe50 | | |
|--------|------------|------------|------------|------------|------------|------------|
| | ACC ↑ | FM ↓ | LA ↑ | ACC ↑ | FM ↓ | LA ↑ |
| GEM | 56.90±0.91 | 5.32±0.86 | 60.12±0.98 | 41.50±0.84 | 5.78±1.20 | 44.24±1.58 |
| ER-Ring | 54.22±0.82 | 10.50±0.63 | 63.92±0.92 | 45.11±2.15 | 8.82±0.52 | 50.73±1.81 |
| MIR | 54.36±1.20 | 7.28±0.72 | 60.25±0.99 | 45.60±1.67 | 5.24±1.72 | 48.00±0.55 |
| CTN | 66.70±1.98 | 4.30±1.94 | 68.02±0.42 | 54.40±1.37 | 5.18±1.61 | 55.40±1.47 |
| DualNet | 72.40±0.54 | **4.04±0.61** | 74.16±0.47 | 57.64±1.36 | 4.43±0.82 | 58.86±0.66 |
| ER-Ring w/ MF | 63.00±2.87 | 13.44±2.82 | 74.40±0.82 | 50.56±2.88 | 15.30±3.34 | 63.76±0.71 |
| CTN w/ MF | 70.30±0.61 | 6.52±0.92 | 74.74±0.73 | 55.70±1.54 | 9.67±1.52 | 61.77±2.20 |
| DualNet w/ MF | 73.34±0.89 | 4.06±0.61 | 74.82±1.42 | 59.40±1.31 | 5.56±1.62 | 62.72±0.65 |
| MuFAN | **75.40±0.44** | 4.40±0.30 | **76.87±1.66** | **67.30±1.57** | 4.38±0.92 | **67.74±1.85** |

We split the minibatch of the feature maps $a \in \mathbb{R}^{B \times C \times W \times H}$ into halves along the channel dimension: $a_1 \in \mathbb{R}^{B \times \frac{C}{2} \times W \times H}$ and $a_2 \in \mathbb{R}^{B \times \frac{C}{2} \times W \times H}$. The first feature map $a_1$ is fed into BN which can learn and preserve a global context as follows:

$$\hat{a}_1 = \gamma_{\text{BN}} \left( \frac{a_1 - \mu_{\text{BN}}}{\sqrt{\sigma_{\text{BN}}^2 + \epsilon}} \right) + \beta_{\text{BN}}, \tag{5}$$

where $\mu_{\text{BN}} = \frac{1}{BHW} \sum_{b=1}^{B} \sum_{w=1}^{W} \sum_{h=1}^{H} a_1$ and $\sigma_{\text{BN}}^2 = \frac{1}{BHW} \sum_{b=1}^{B} \sum_{w=1}^{W} \sum_{h=1}^{H} (a_1 - \mu_{\text{BN}})^2$ are the mean and variance calculated along the minibatch dimension. $\gamma_{\text{BN}}$ and $\beta_{\text{BN}}$ are the affine transformation parameters of BN. The second feature map $a_2$ undergoes the combination of IN and LN which utilize instance-specific running statistics along the spatial dimension as follows:

$$\hat{a}_2 = \gamma_{\text{IN,LN}} \left( \frac{a_2 - \sum_{k \in \{\text{IN,LN}\}} w_k \mu_k}{\sqrt{\sum_{k \in \{\text{IN,LN}\}} w'_k \sigma_k^2 + \epsilon}} \right) + \beta_{\text{IN,LN}}, \tag{6}$$

where $\mu_{\text{IN}} = \frac{1}{HW} \sum_{w=1}^{W} \sum_{h=1}^{H} a_2$ and $\sigma_{\text{IN}}^2 = \frac{1}{HW} \sum_{w=1}^{W} \sum_{h=1}^{H} (a_2 - \mu_{\text{IN}})^2$ are the mean and variance calculated along the channel dimension, and $\mu_{\text{LN}} = \frac{1}{CHW} \sum_{c=1}^{C} \sum_{w=1}^{W} \sum_{h=1}^{H} a_2$ and $\sigma_{\text{LN}}^2 = \frac{1}{CHW} \sum_{c=1}^{C} \sum_{w=1}^{W} \sum_{h=1}^{H} (a_2 - \mu_{\text{LN}})^2$ are the mean and variance calculated along the layer dimension. $w$ and $w'$ are balancing weights between IN and LN which are optimized by backpropagation. Using a softmax, we make $\sum_{k \in \{\text{IN,LN}\}} w_k = 1$ and $\sum_{k \in \{\text{IN,LN}\}} w'_k = 1$. $\gamma_{\text{IN,LN}}$ and $\beta_{\text{IN,LN}}$ are the affine transformation parameters of the combination of IN and LN. Then, the concatenation of $\hat{a}_1$ and $\hat{a}_2$ along the channel dimension is fed into the next convolutional kernels. As shown in Huang et al. (2020) and Luo et al. (2018), depending on the characteristics of data or task, the spatial normalization layer that exhibits stable performance is different. Thus, we utilize the combination of IN and LN for robustness in various datasets.

BN commonly performs better than spatial normalization layers in image classification tasks because it is beneficial in stabilizing training and improving generalization. However, due to the nature of CL, the basic assumption of BN that the running statistics between training and inference are consistent cannot be held. This inconsistency causes negative bias in moments and provokes catastrophic forgetting. In contrast, spatial normalization layers, such as IN, GN, and LN, compute the running statistics along the channel or layer dimension. Therefore, these normalization layers are effective in reducing the negative bias in inference because they do not require the minibatch statistics obtained

Table 2: Comparison results in a task-free setting where the notion of tasks are unavailable ($S$: the criterion of a new potential task in Appendix F).

| | Online Task-free CL | | |
|---|---|---|---|
| ACC ↑ | CIFAR100 | miniIMN | CORe50 |
| ER | 20.5±0.9 | 11.0±0.5 | 28.2±1.7 |
| DER++ | 20.7±2.7 | 13.7±1.2 | 31.7±4.7 |
| DualNet | 25.5±0.7 | 20.9±1.6 | 35.6±0.6 |
| MuFAN ($S = 5$) | **39.6±0.3** | **34.7±2.1** | 47.2±3.6 |
| MuFAN ($S = 10$) | 38.2±0.8 | 33.3±1.5 | **48.5±1.8** |

Table 3: Effectiveness of multi-scale feature maps (ST: standard, BU: bottom-up, and TD: top-down).

| | ACC ↑ | SVHN | CIFAR100 |
|---|---|---|---|
| 1) | ER-Ring (Fixed, ST) | 63.6±0.5 | 61.9±0.2 |
| | ER-Ring (Fixed, BU) | 65.8±1.3 | 64.0±0.4 |
| 2) | ER-Ring (ST) | 83.7±4.2 | 62.8±0.9 |
| | ER-Ring (BU) | 78.3±2.3 | 55.0±0.9 |
| 3) | ER-Ring (TD) | **92.3±0.3** | **69.3±1.6** |

during training. Unlike SN and CN which merge normalization for stability and normalization for plasticity in a mixture or sequential way, SPN applies independent normalization on the divided feature maps. We suggest this structure to allow the convolutional kernels to independently utilize differently normalized feature maps during training. Specifically, the channel-wise separation boosts both the stable learning property of BN and the forgetting-prevention ability of IN and LN combinations. We confirmed that the proposed SPN outperforms SN and CN through experiments.

# 4 EXPERIMENTS

## 4.1 EXPERIMENTAL SETUP

We tested the effectiveness of MuFAN on online task-incremental and online task-free CL settings. We compare MuFAN with the following state-of-the-art baselines: GEM (Lopez-Paz & Ranzato, 2017a), ER-Ring (Chaudhry et al., 2019), DER++ (Buzzega et al., 2020), MIR (Aljundi et al., 2019), CTN (Pham et al., 2020), and DualNet (Pham et al., 2021a). We reported the results of five runs for all the experiments. We used three standard metrics: the average accuracy (ACC ↑) (Lopez-Paz et al., 2017), forgetting measure (FM ↓) (Chaudhry et al., 2018a), and learning accuracy (LA ↑) (Riemer et al., 2018), to measure accuracy, stability (FM) and plasticity (LA). Further details on the evaluation metrics, baselines, training, data augmentation, and hyperparameters can be found in Appendix D and E. We observed MuFan's insensitivity to hyperparameter tuning. Although we used a single hyperparameter configuration that gave the best average performance over all four benchmarks, MuFAN still outperformed the baselines (Appendix E has more results on hyperparameter tuning).

**Datasets.** We consider four CL benchmarks in our experiments. For Split SVHN (Tang & Matteson, 2020), we split the SVHN dataset (Netzer et al., 2011) into 5 tasks, each task containing 2 different classes sampled without replacement from a total of 10 classes. Similarly, for Split CIFAR100 (Lopez-Paz & Ranzato, 2017a), we split the CIFAR100 dataset (Krizhevsky et al., 2009) into 20 tasks, each task containing 5 different classes. The Split miniImageNet (Split miniIMN) (Chaudhry et al., 2018b) is constructed by splitting the miniImageNet dataset (Vinyals et al., 2016) into 20 disjoint tasks. Lastly, the CORe50 benchmark is constructed from the original CORe50 dataset (Lomonaco & Maltoni, 2017), with a sequence of 10 tasks.

**Architecture.** As the pre-trained encoder $e$, we used an ImageNet-pretrained EfficientNet-lite0 (Tan & Le, 2019) on Split SVHN, Split CIFAR100, and CORe50 and a COCO-pretrained SSDlite (Howard et al., 2019), which uses MobileNetV3 as a backbone, on Split miniIMN. We implemented the SSDlite model trained on the MS COCO object detection dataset (Lin et al., 2014) from scratch. For the classifier $m$, we used a randomly initialized ResNet18 (He et al., 2016). Details on the libraries used can be found in Appendix E.

## 4.2 RESULTS ON ONLINE CONTINUAL LEARNING BENCHMARKS

**Task-incremental setting.** Table 1 shows the performance comparison on four CL benchmarks. Overall, GEM showed a lower ACC than the rehearsal-based methods, ER-Ring and MIR. DualNet showed a significantly improved performance, particularly, on Split CIFAR100 and Split miniIMN. However, MuFAN exhibited superiority over the other state-of-the-art methods on all four benchmarks. Specifically, on CORe50, MuFAN exhibited 17% and 15% relative improvements in ACC and LA over DualNet, respectively. Despite the trade-off between FM and LA, MuFAN obtained the highest LA and comparable or lowest FM, resulting in showing the highest ACC on all four benchmarks. In other words, MuFAN succeeded in addressing the stability-plasticity dilemma in online CL. Furthermore, to test the scalability of the aggregated multi-scale feature map from the pre-trained model, we

Table 4: Importance of a distance metric used in $\mathcal{L}_{\text{D-CSD}}$.

| ER-Ring | Split CIFAR100 | | Split miniIMN | |
|---|---|---|---|---|
| | ACC ↑ | FM ↓ | ACC ↑ | FM ↓ |
| L2 Norm | 64.5±0.8 | 4.7±1.2 | 60.3±1.7 | 4.7±1.0 |
| CS | **64.8±0.5** | **4.6±0.3** | **60.9±1.0** | **4.5±0.6** |
| ADC | 64.3±0.9 | 4.9±0.8 | 60.3±0.7 | 4.9±0.5 |

Table 5: Ablation study on a base task in the structure-wise distillation loss.

| | CORe50 | | |
|---|---|---|---|
| | ACC ↑ | FM ↓ | LA ↑ |
| $\mathcal{L}_{\text{D-CSD}}$ | **67.30±1.57** | **4.38±0.92** | 67.74±1.85 |
| $\mathcal{L}_{\text{D-FSD}}$ | 66.54±1.22 | 5.98±1.63 | 68.25±1.66 |
| $\mathcal{L}_{\text{D-LSD}}$ | 67.09±1.89 | 5.42±2.01 | **68.30±1.22** |

Table 6: Comparison results of different normalization layers. The same backbone and 50 memory slots per task are used by all methods.

| Norm. | BN | | | RBN | | | IN | | | GN | | | LN | | | SN | | | CN | | | SPN (Ours) | | |
|---|---|---|---|---|---|---|---|---|---|---|---|---|---|---|---|---|---|---|---|---|---|---|---|---|
| DER++ | ACC | FM | LA | ACC | FM | LA | ACC | FM | LA | ACC | FM | LA | ACC | FM | LA | ACC | FM | LA | ACC | FM | LA | ACC | FM | LA |
| miniIMN | 58.9 | 7.7 | 63.8 | 62.2 | 7.4 | 67.5 | 52.9 | 9.7 | 58.5 | 59.9 | 4.3 | 59.6 | 54.8 | **3.9** | 53.6 | 62.4 | 8.0 | 66.8 | 63.5 | 9.0 | 70.8 | **64.8** | 9.3 | **72.2** |
| CORe50 | 45.6 | 7.2 | 49.0 | 49.6 | 8.9 | 55.7 | 42.0 | **3.3** | 40.8 | 42.3 | 7.9 | 46.3 | 44.9 | 6.3 | 47.2 | 52.4 | 8.6 | 58.9 | 48.6 | 7.7 | 54.1 | **55.0** | 9.0 | **62.1** |

integrated it into the baselines and reported the results in Table 1. MuFAN still outperformed the baselines using the multi-scale feature map. We observed a significant increase in LA in ER-Ring and CTN. However, relatively, the multi-scale feature map was less effective for DualNet. This is because that DualNet already leverages general representations using self-supervised learning. Consequently, we demonstrated that our proposed concept of leveraging the aggregated multi-scale feature map is scalable with existing online CL methods. To check the effectiveness of the proposed components more clearly, we first tested each component with the baseline CL method in several experiments.

**Task-free setting.** To test the effectiveness of MuFAN in an online task-free CL setting where the notion of tasks and task boundaries are unavailable during training and inference, we revised the original Eq. 2 to Eq. 11 in Appendix F. Table 2 presents the comparison results on three benchmarks with two different $S$s (the criterion of a new potential task). We confirmed that MuFAN is still superior to the baselines in a task-free setting. Appendix F covers more details.

**Multi-scale feature maps from a pre-trained encoder.** In Table 3, to demonstrate the significance of the top-down module, we ablated over three scenarios with standard, bottom-up, and top-down modules. A standard module represents a case that only the last convolutional feature map from the pre-trained network is utilized. For the standard and bottom-up modules, due to the small height and width size, they are applicable only when the pre-trained network is used as a feature extractor. The first and second scenarios are that the pre-trained model is used as a feature extractor. The difference between the two is whether the feature extractor is fixed during training. The third is that the pre-trained model is used as an encoder. As shown in the results of Table 3, using the top-down module with the pre-trained encoder exhibits superiority over all cases of using the standard or bottom-up module. In addition, we tested which and how many layers must be used to make the most informative multi-scale feature map and reported the results in Table 7. Rather than only the first layer or last three layers, the four layers including a shallow layer showed impressive results. We also performed ablation studies by varying pre-trained models, including a self-supervised model, as the encoder $e$ and checked that MuFAN mostly shows better performance than DualNet with any pre-trained model (Appendix G).

Table 7: Ablation study on the layer and the number of layers that must be used.

| ACC ↑ | CIFAR100 | CORe50 |
|---|---|---|
| First layer | 73.89±1.94 | 62.01±3.04 |
| 3 Layers | 72.22±1.03 | 61.07±0.99 |
| 4 Layers | **75.86±0.35** | **67.30±1.57** |

**Model complexity.** We measured and reported the complexity of MuFAN and the baselines in Table 8. Because MuFAN uses EfficientNet-lite0, which is a lightweight version of EfficientNet, as the pre-trained encoder, only minimal parameters are added to the ResNet18 backbone. Compared with the baselines, and especially with the recent state-of-the-art method, DualNet, the increase in the number of parameters is not significant. Also, the total number of hyperparameters of MuFAN is smaller than those of DualNet and CTN. Further analysis of model complexity in terms of FLOP and memory is provided in Appendix H.

Table 8: Model complexity.

| | # Params |
|---|---|
| ER-Ring | 11,202,162 |
| CTN | 11,261,362 |
| DualNet | 13,272,690 |
| MuFAN | 14,565,234 |

**Cross-task structure-wise distillation ($\mathcal{L}_{\text{D-CSD}}$).** First, to evaluate whether the intuition of our proposed structure-wise distillation loss indeed holds, we visualized the latent space of MuFAN using T-SNE (Van der Maaten & Hinton, 2008) as shown in Appendix I (Figure 4). We observed that the relation between cross-task data points is well maintained throughout the stream of tasks. In addition to CS, we tested other distance metrics, Euclidean distance (2-norm distance, L2 norm) and angular distance using arccos (ADA) (Cer et al., 2018), to check whether there is a better way

Table 9: Ablation study on the proposed three components and data augmentation (DA).

| | | | | | Split miniIMN | | | CORe50 | | |
|---|---|---|---|---|---|---|---|---|---|---|
| | | | | | ACC ↑ | FM ↓ | LA ↑ | ACC ↑ | FM ↓ | LA ↑ |
| $\mathcal{L}_{\text{CE}} + \mathcal{L}_{\text{D-ER}} + \lambda_{\text{D-CTN}}\mathcal{L}_{\text{D-CTN}}$ | | | | | 61.56±0.70 | 3.95±0.67 | 61.44±1.31 | 48.71±0.91 | 7.12±0.81 | 52.55±0.98 |
| | MF | DA | $\mathcal{L}_{\text{D-CSD}}$ | SPN | | | | | | |
| (1) | ✓ | | | | 72.08±0.42 | 5.72±0.42 | 76.18±0.68 | 56.00±1.63 | 11.52±0.88 | 65.30±2.14 |
| (2) | | ✓ | | | 62.34±0.91 | 3.70±0.49 | 60.57±0.88 | 51.22±0.68 | 5.29±0.49 | 52.14±0.78 |
| (3) | | | ✓ | | 63.71±0.59 | 3.47±0.41 | 62.48±0.32 | 51.88±1.11 | 4.60±0.37 | 52.63±1.03 |
| (4) | | | | ✓ | 65.62±0.58 | 5.29±0.56 | 65.15±0.99 | 56.03±1.81 | 4.39±1.91 | 57.31±1.62 |
| (5) | ✓ | ✓ | | | 73.22±0.90 | 5.10±0.66 | 75.50±0.69 | 60.36±1.27 | 7.76±1.05 | 65.90±1.48 |
| (6) | ✓ | | ✓ | | 73.36±0.58 | 3.60±0.36 | 75.46±1.20 | 60.66±0.76 | 5.02±0.82 | 64.24±1.15 |
| (7) | ✓ | | | ✓ | 75.52±0.88 | 5.90±0.38 | **77.80±0.79** | 61.57±2.16 | 8.52±2.72 | **68.62±1.50** |
| (8) | ✓ | ✓ | ✓ | | 74.14±0.81 | **3.34±0.79** | 74.42±0.86 | 62.98±1.34 | **3.92±0.68** | 63.44±1.64 |
| (9) | ✓ | ✓ | | ✓ | 75.10±0.92 | 5.65±0.80 | 77.66±0.48 | 65.19±1.05 | 6.88±0.62 | 68.31±1.82 |
| (10) | ✓ | | ✓ | ✓ | 75.11±1.16 | 4.44±0.89 | 76.61±1.46 | 66.00±1.81 | 5.46±0.80 | 68.73±1.50 |
| (11) | ✓ | ✓ | ✓ | ✓ | **75.40±0.44** | 4.40±0.30 | 76.87±1.66 | **67.30±1.57** | 4.38±0.92 | 67.74±1.85 |

to penalize the distance between samples across tasks. As shown in Table 4, although the result using CS is slightly superior, in general, the proposed structure-wise distillation loss showed stable performance whether using a distance-based metric (Eq. 13 in Appendix J) or any other angle-based metric (Eq. 15 in Appendix J). Besides, we tested the effectiveness of suppressing forgetting when the first task (Eq. 16 in Appendix K) or the most recently learned task (Eq. 17 in Appendix K) is used instead of the consecutive tasks to build the relation and reported the results in Table 5. Among the three losses, $\mathcal{L}_{\text{D-CSD}}$ brought most impressive improvement in FM. As a result, we observed that using consecutive tasks (Eq. 2) is most effective in ACC and FM. Lastly, we assessed the number of samples per task $N$ that must be used to efficiently utilize $\mathcal{L}_{\text{D-CSD}}$. We confirmed that since $N$ is equal to the number of classes in each task, it is possible to build the reliable relation across tasks for distillation. Appendix K covers more details of equations and results.

**Stability-plasticity normalization.** As shown in Table 6, our SPN exhibited superiority over both single and multiple normalization layers, especially on CORe50. SPN showed a 13% relative improvement in ACC over CN. More precisely, RBN, a variant of BN, had the best performance among the single normalization layers. RBN is advantageous in the nature of CL because it additionally leverages the statistics of instance-specific features for feature calibration. Although it tends to vary according to the dataset, the spatial normalization layers such as IN, GN, and LN show low LA but also low FM. Among IN, GN, and LN, we only utilize IN and LN, not GN, for SPN to avoid an additional hyperparameter (the number of groups). We also tested the performance of SPN and the other normalization layers on ER-Ring and obtained consistent results (Table 19 in Appendix L). As an ablation study, we conducted the experiments of using only IN or LN for the stability normalization operation and summarized the results with analysis in Appendix L.

**Ablation study of each component.** We assessed the importance of each component proposed in this study by using various combinations of the components. As shown in Table 9, we verified that the aggregated multi-scale feature map and SPN consistently enhance plasticity (LA ↑), and the structure-wise distillation loss enhances stability (FM ↓) under diverse combinations. Most surprisingly, the LA results of entry (1) in Table 9 verify that multi-scale semantic feature maps generate a very strong training signal that can help to quickly acquire new information on the fly. The results of entries (1) and (6) in Table 9 validated that catastrophic forgetting can be largely suppressed by a relational distillation signal from our structure-wise loss while maintaining plasticity.

## 5 CONCLUSION

For online CL, wherein data comes strictly in a streaming manner, achieving high plasticity is challenging; thus, it is difficult to expect a significant final performance by only improving stability. In this study, we propose a novel online CL framework named MuFAN, which enhances both stability and plasticity while being less impediment to one another. This is the first CL method that leverages multi-scale feature maps, which are constructed by projecting raw images into meaningful spaces, as the input for the classifier in order to improve plasticity. In addition, we propose a novel structure-wise distillation loss that alleviates forgetting using a relation between samples across tasks. Finally, we present SPN that can be easily integrated into online CL methods to maintain high plasticity and stability simultaneously. Carefully designed ablation studies demonstrated the significance and scalability of the proposed three components in MuFAN. In future work, we would like to delve into the utility of pre-trained models in a few-shot CL setting.

ACKNOWLEDGEMENT

This work was supported by the National Research Foundation of Korea (NRF) grant funded by the Korea government (MSIT) [2022R1A3B1077720], Institute of Information & Communications Technology Planning & Evaluation (IITP) grant funded by the Korea government (MSIT) [2021-0-01343, AI Graduate School Program (SNU); 2022-0-00959; 2022-00155966, AI Convergence Innovation Human Resources Development (EWU)], Samsung Electronics (IO221213-04119-01), and LG Innotek.

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

# A   ONLINE CL AND OFFLINE CL

In the conventional scenario of online learning, they assume data that comes strictly in a streaming manner. That is, the model immediately adapts to data at that moment, and the streaming instance or small batch is immediately discarded (Polikar et al., 2001). Therefore, storing all of the streaming data in the online validation buffer during a short moment suggested in the papers (Hu et al., 2022; Aljundi et al., 2019) is not common in the conventional online learning scenario. However, we tested whether equaling the number of iterations and the number of epochs brings similar performance results. As shown in Table 10, even if the number of iterations and the number of epochs match one another, we still observed lower LA and lower FM in online CL. In other words, the characteristics of online CL are maintained unless offline storage is assumed.

Table 10: Comparison on equaling the number of epochs (10 epochs) and the number of iterations (10 iterations).

| Method | Split CIFAR100 | | | Split miniIMN | | | CORe50 | | |
|---|---|---|---|---|---|---|---|---|---|
| | ACC ↑ | FM ↓ | LA ↑ | ACC ↑ | FM ↓ | LA ↑ | ACC ↑ | FM ↓ | LA ↑ |
| 10 Epochs | 60.38 | 17.67 | 77.16 | 59.14 | 20.02 | 78.10 | 53.68 | 11.76 | 65.32 |
| 10 Iterations | 58.93 | 13.40 | 71.62 | 58.70 | 9.87 | 67.13 | 51.70 | 9.30 | 59.05 |

# B   NORMALIZATION LAYERS

Batch normalization (BN) (Ioffe & Szegedy, 2015) has been demonstrated beneficial in a wide range of deep learning applications. Inspired by BN, varying normalization layers have been proposed by exploring different normalization dimensions. According to that how the running statistics are calculated along which convolutional feature dimensions, we categorize normalization layers into three approaches: 1) BN and its variants, batch renormalization (BRN) (Ioffe, 2017) and representative batch Normalization (RBN) (Gao et al., 2021), 2) instance normalization (IN) (Ulyanov et al., 2016), layer normalization (LN) (Ba et al., 2016), and group normalization (GN) (Wu & He, 2018), and 3) switchable normalization (SN) (Luo et al., 2018) and continual normalization (CN) (Pham et al., 2021b).

## B.1   BN AND ITS VARIANTS, BRN AND RBN

When a convolutional feature map is fed into a BN (Ioffe & Szegedy, 2015) layer, the mean and variance of a current minibatch are calculated to perform normalization. A BN layer then goes through an affine transformation process of scaling and shifting the normalized features with two learnable parameters.

BN can promote preserving the global context of a task during training. However, BN tends to fail to show full performance when a minibatch is small or non-independent and identical distribution (non-i.i.d.). To address this problem, BRN (Ioffe, 2017) proposed a simple extension of BN, which implements a re-parametrization trick before the affine transformation.

Recently, RBN (Gao et al., 2021) proposed another simple extension of BN that goes through an additional centering and scaling calibration process by leveraging the statistics of instance-specific features to alleviate the inconsistency of the running moments between training and inference. Therefore, RBN helps construct a stable feature distribution between channels with minimal additional cost.

## B.2   IN, LN, AND GN

The works in the second approach normalize a convolutional feature map along the spatial (channel or layer) dimension, which is advantageous to the discrepancy between training and inference.

IN (Ulyanov et al., 2016) was first proposed for an image style transfer network. The mean and variance are calculated for each channel of a single input feature. Similar to IN, LN (Ba et al., 2016),

which is widely used in recurrent neural networks, normalizes a convolutional feature map along the layer dimension instead of the minibatch dimension. In other words, LN calculates the mean and variance along all channels for an input feature. Lastly, GN (Wu & He, 2018) is closely related to IN and LN, but instead of normalizing for each individual channel or along all channels; GN normalizes an input feature by dividing the channels into $G$ groups. Therefore, when $G = 1$, GN behaves the same as LN, and when $G$ equals the number of channels, GN behaves the same as IN.

### B.3 SN AND CN

Third, SN and CN normalize a convolutional feature map along both the minibatch and spatial dimensions to take advantage of both approaches.

In SN, operations in BN, IN, and LN are used to compute the means and variances of three types estimated along the minibatch, channel, and layer dimensions. By learning their blending weights, SN combines the statistics of these three normalization layers for final normalization.

Moreover, recently, CN (Pham et al., 2021b) was specially designed to address the non-i.i.d. nature of CL. To mitigate the side effects arising from the model normalizing previous tasks' data using the moments of a current task during inference, they first normalize an input feature along the spatial dimension (GN) without the affine transformation. Then, the spatially normalized feature map is fed into a normalization operation along the minibatch dimension (BN). Mathematically, CN is defined as follows:

$$\boldsymbol{a}_{\text{GN}} \leftarrow \text{GN}_{1,0}(\boldsymbol{a}); \quad \boldsymbol{a}_{\text{CN}} \leftarrow \boldsymbol{\gamma}\text{BN}_{1,0}(\boldsymbol{a}_{\text{GN}}) + \boldsymbol{\beta}, \tag{7}$$

where $\text{GN}_{1,0}$ and $\text{BN}_{1,0}$ represent a group normalization layer and batch normalization layer without an affine transform, respectively. Empirically, CN exhibited improvements in the final performance. However, still, it is hard to argue that the benefit of BN was fully exploited in CN due to the characteristics of calculation in order. We, thus, propose the SPN module that performs different normalization operations in a parallel way.

## C MIXING STRATEGIES

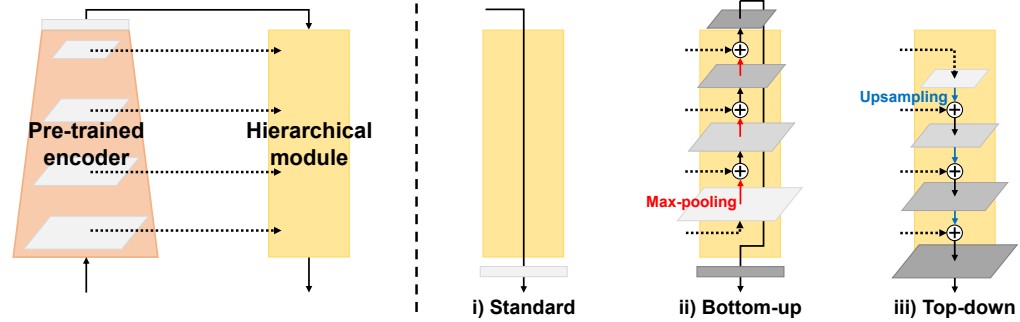

Figure 3: Hierarchical module. For the standard and bottom-up modules, the final feature map is fed into the classification layer. For the top-down module, the final feature map is fed into the entire classifier.

Features of different scales extracted by a deep neural network have different characteristics and the utility of multi-scale features has been exploited in several fields. Starting with a U-Net (Ronneberger et al., 2015), methods for augmenting multi-scale features have been proposed (Lin et al., 2019; Guo et al., 2020; He et al., 2019; Sauer et al., 2021). As described in Figure 3, broadly, these methods can be divided into two major categories: top-down and bottom-up approaches. The top-down method propagates the features from deeper layers to shallow layers, whereas the bottom-up approach propagates the features from shallow layers to deeper layers. In the case of MuFAN, it follows the proposed method by Sauer et al. (2021), which is based on fixed, differentiable random projections.

Sauer et al. (2021) proposed an augmenting method that consists of two different strategies to efficiently mix features from different layers: 1) Cross-Channel Mixing (CCM), and 2) Cross-Scale

Mixing (CSM). The purpose of implementing CCM is to mix across channels, which encourages information preservation of each feature by permutation (Kingma & Dhariwal, 2018). CCM utilizes a randomly initialized $1 \times 1$ convolution with an equal number of output and input channels. Second, CSM mixes features from two different scales. For the top-down module, to match the resolution and the number of channels of shallow layers, the feature from deeper layers enlarges a resolution by bilinear upsampling and reduces the number of channels by a random $3 \times 3$ convolution. For the bottom-up module, oppositely, the feature from shallow layers shrinks a resolution by max-pooling and increases the number of channels by a random $3 \times 3$ convolution. Then, the resized feature is added to another feature from a different scale element-wisely, as illustrated in Figure 3. For detailed intuition and analysis about the mixing strategies, please refer to Sauer et al. (2021).

By considering the resolution of a final multi-scale feature map, the output from the bottom-up module is fed into a classification layer without an average pooling.

Table 11: Ablation study on the mixing strategies (CCM: Cross-Channel Mixing, and CSM: Cross-Scale Mixing).

| ACC ↑ | CIFAR100 | CORe50 |
|---|---|---|
| CSM | 75.09±0.50 | 64.78±1.31 |
| CCM + CSM | 75.86±0.35 | 67.30±1.57 |

For MuFAN, we conducted an ablation study on the mixing strategies (CCM and CSM) proposed by Sauer et al. (2021). To test whether CCM is beneficial for MuFAN, we checked the performance difference according to using only CSM and both. As shown in Table 11, we observed that using both mixing strategies is effective for MuFAN.

## D    EVALUATION METRICS

Three common ongoing learning metrics—average accuracy (ACC) (Lopez-Paz et al., 2017), forgetting measure (FM) (Chaudhry et al., 2018a), and learning accuracy (LA) (Riemer et al., 2018)—are used to assess performance.

For notation, the accuracy of the model, as measured on the test set $\mathcal{D}_j^{te}$ after it has been trained up to the most recent dataset $\mathcal{D}_i$ of task $i$, is denoted by the symbol $a_{i,j}$.

- ACC (Higher is better): evaluates the final performance of a continual learning method in terms of accuracy.

$$\text{ACC}(\uparrow) = \frac{1}{T} \sum_{j=1}^{T} a_{T,j}. \tag{8}$$

- FM (Lower is better): evaluates the performance of a continual learning method in terms of stability.

$$\text{FM}(\downarrow) = \frac{1}{T-1} \sum_{j=1}^{T-1} \max_{l \in \{1, \cdots, T-1\}} a_{l,j} - a_{T,j}. \tag{9}$$

- LA (Higher is better): evaluates the performance of a continual learning method in terms of plasticity.

$$\text{LA}(\uparrow) = \frac{1}{T} \sum_{j=1}^{T} a_{j,j}. \tag{10}$$

# E    EXPERIMENT DETAILS

We conducted all experiments using the Pytorch (Paszke et al., 2019) framework on a single RTX8000 GPU. For varying pre-trained models, we implemented open python libraries. For the EfficientNet-lite (Tan & Le, 2019) and ResNet18 (He et al., 2016) trained by the ImageNet (Deng et al., 2009) dataset in a supervised manner, we utilized the *timm* library. For the ResNet50 trained by the ImageNet dataset in a self-supervised manner, we utilized the *vissl* library. Lastly, for the SSDlite (Howard et al., 2019) trained by the MS COCO object detection dataset (Lin et al., 2014), we utilized the Pytorch *torchvision* library.

For all experiments, we run each experiment five times using different initialization with the same sequence of tasks.

## E.1    DATASET SUMMARY

We summarize the datasets used in our experiments in Table 12. We applied the same transformations on MuFAN and all baselines except for DualNet. Because DualNet utilizes self-supervised learning, it contains random cropping, resizing, horizontal flipping, color jittering, and so on. In the case of MuFAN and remaining baselines, we utilized random cropping, horizontal flip, and resizing. As an exception, for Split SVHN, because data is sensitive to horizontal flip, we did not utilize it.

One difference is that we utilized data augmentation only on data from the replay buffer. That is, we applied random cropping, horizontal flip, and resizing to data from the replay buffer and only resizing to streaming data. This is because we observed an increase in the final performance when data augmentation is implemented only to data from the replay buffer in general.

Table 12: Dataset summary, including a number of classes, a number of tasks, a number of data, and dimensions with data augmentation.

| Dataset | Classes | # Tasks | Train | Test | Dim. |
|---|---|---|---|---|---|
| Split SVHN | 10 | 5 | 73,257 | 26,032 | $3 \times 32 \times 32$ |
| Split CIFAR100 | 100 | 20 | 50,000 | $10,000 \times 32$ | $3 \times 32 \times 32$ |
| Split miniIMN | 100 | 20 | 50,000 | $10,000 \times 84$ | $3 \times 128 \times 128$ |
| CORe50 | 50 | 10 | 119,984 | 44,971 | $3 \times 128 \times 128$ |

## E.2    BASELINES

A category of CL can be divided into three families: a regularization-based approach, a dynamic architecture approach, and a replay-based approach. First, the regularization-based approach aims at consolidating old knowledge when learning new tasks, commonly without relying on an experience replay buffer (Li & Hoiem, 2017; Kirkpatrick et al., 2017; Lee et al., 2017). For example, LwF (Li & Hoiem, 2017) hinders forgetting by using outputs from a previous model and EWC (Kirkpatrick et al., 2017) suppresses the changes in network parameters estimated as important. Second, the dynamic architecture approach fixes or/and expands network architecture and parameters to achieve both high stability and plasticity in a less limited manner (Rusu et al., 2016; Mallya & Lazebnik, 2018; Serra et al., 2018). Lastly, the replay-based approach stores a limited number of examples in previous tasks and replays them during the training of new tasks (Chaudhry et al., 2019; Lopez-Paz & Ranzato, 2017a; Chaudhry et al., 2018b; Rebuffi et al., 2017; Shin et al., 2017). This approach regularizes forgetting either by forcing output activations of the stored individual data represented by the current model and that represented by the previous model to be the same (Chaudhry et al., 2019) or by projecting the gradient direction on the current task outlined by the stored data (Lopez-Paz & Ranzato, 2017a).

MuFAN is also included in the replay-based approach. Thus, in this study, we compared MuFAN with several baselines in the replay-based approach. A brief description of the baseline (competitor) is provided as follows:

- GEM (Lopez-Paz & Ranzato, 2017a): regularizes forgetting by projecting the gradient direction on the current task outlined by the stored data.

- AGEM (Chaudhry et al., 2018b): an efficient version of GEM, which relaxes the gradient direction as one direction.
- MER (Riemer et al., 2018): uses a variant of the Reptile approach and optimizes the gradient inner product between each sample pair in the buffer.
- ER-Ring (Chaudhry et al., 2019): a standard experience replay that stores a subset of data from previous tasks and optimizes a multitask loss.
- DER (Buzzega et al., 2020): applies a $\ell_2$ loss to regularize the difference between current and past logits.
- DER++ (Buzzega et al., 2020): a variant of DER with an additional $\ell_2$ regularizer between the current logit and ground-truth label.
- MIR (Aljundi et al., 2019): a variant of ER that, by sorting, chooses the samples that increase the model's forgetting.
- CTN (Pham et al., 2020): models both common and task-specific features via a lightweight controller module and bilevel optimization objective.
- DualNet (Pham et al., 2021a): Inspired by complementary learning systems, consists of fast and slow networks, which focus on capturing new knowledge and learning a general representation using self-supervised learning, respectively.

Due to the page limit, we included the results of AGEM and MER showing similar performance to GEM in Appendix. Overall, it showed lower or comparable performance over the replay-based methods, ER-Ring and MIR.

Table 13: Comparison results of AGEM and MER on four CL benchmarks. The same backbone and 50 memory slots per task are used by all methods.

| Method | Split SVHN | | | Split CIFAR100 | | |
|---|---|---|---|---|---|---|
| | ACC ↑ | FM ↓ | LA ↑ | ACC ↑ | FM ↓ | LA ↑ |
| AGEM | 83.48±2.54 | 15.10±3.39 | 95.58±0.33 | 57.77±0.97 | 6.08±0.82 | 64.22±1.13 |
| MER | 89.45±1.21 | 4.55±1.49 | 92.55±1.21 | 59.57±1.12 | 9.44±0.91 | 69.83±0.79 |

| Method | Split miniIMN | | | CORe50 | | |
|---|---|---|---|---|---|---|
| | ACC ↑ | FM ↓ | LA ↑ | ACC ↑ | FM ↓ | LA ↑ |
| AGEM | 54.92±1.32 | 6.42±0.77 | 60.61±1.11 | 40.62±1.84 | 10.58±1.28 | 47.74±0.95 |
| MER | 58.48±0.91 | 6.01±1.19 | 64.12±0.62 | 40.10±0.78 | 7.06±0.68 | 46.66±1.02 |

### E.3 HYPERPARAMETER SELECTION

By following the online task-incremental setting used in Pham et al. (2021a), all methods are trained over one epoch with a minibatch size of 10 on Split SVHN, Split CIFAR100, and Split miniIMN and 32 on CORe50. We store 50 samples per task in the replay buffer and use a ring-buffer management strategy (Lopez-Paz & Ranzato, 2017b). We provide the hyperparameters of each method.

- GEM
  - Learning rate: 0.03 (Split SVHN, Split CIFAR100), 0.05 (Split miniIMN)
  - Gradient noise $\gamma$: 0.5 (all experiments)
  - Number of gradient updates: 1 (Split CIFAR100, Split miniIMN), 2 (Split SVHN)
- AGEM
  - Learning rate: 0.03 (Split SVHN), 0.1 (Split CIFAR100), 0.3 (Split miniIMN)
  - Number of to estimate gradient constraints: 850 (all experiments)
  - Number of gradient updates: 1 (Split CIFAR100, Split miniIMN), 2 (Split SVHN)
- MER
  - Learning rate: 0.03 (Split SVHN), 0.05 (Split miniIMN), 0.1 (Split CIFAR100)

- – Replay batch size: 64 (Split SVHN, Split CIFAR100), 128 (Split miniIMN)
- – Number of gradient updates: 2 (Split SVHN), 3 (Split CIFAR100, Split miniIMN)
- – Across batch learning rate $\gamma$: 0.3 (all experiments)

- ER-Ring
  - – Learning rate: 0.03 (all benchmarks)
  - – Replay batch size: 10 (Split SVHN, Split CIFAR100, Split miniIMN), 32 (CORe50)
  - – Number of gradient updates: 2 (all benchmarks)

- DER
  - – Learning rate: 0.03 (CORe50)
  - – Replay batch size: 64 (CORe50)
  - – Number of gradient updates: 2 (CORe50)

- DER++
  - – Learning rate: 0.03 (CORe50)
  - – Replay batch size: 64 (CORe50)
  - – Trade-off strength between soft and hard labels: 1 (CORe50)
  - – Number of gradient updates: 2 (CORe50)

- MIR
  - – Learning rate: 0.03 (all benchmarks)
  - – Replay batch size: 10 (all benchmarks)
  - – Number of gradient updates: 2 (Split SVHN), 3 (Split CIFAR100, Split miniIMN, CORe50)

- CTN
  - – Inner learning rate $\alpha$: 0.01 (all benchmarks)
  - – Outer learning rate $\beta$: 0.05 (all benchmarks)
  - – Replay batch size: 64 (Split SVHN, Split CIFAR100, Split miniIMN), 32 (CORe50)
  - – Regularization strength $\lambda$: 100 (all benchmarks)
  - – Temperature $\tau$: 5 (all benchmarks)
  - – Semantic memory size in percentage of total memory: 20% (all benchmarks)
  - – Embedding dimension of a MLP layer to map the task identifiers: 16 (Split SVHN, CORe50), 64 (Split CIFAR100, Split miniIMN)
  - – Number of inner updates: 2 (all benchmarks)
  - – Number of outer updates: 2 (all benchmarks)

- DualNet
  - – Slow learner's SGD learning rate: 3e-4 (Split SVHN, Split CIFAR100, Split miniIMN), 1e-4 (CORe50)
  - – Slow learner's Look-ahead learning rate: 0.5 (all benchmarks)
  - – Fast learner's learning rate: 0.03 (all benchmarks)
  - – Replay batch size: 10 (Split SVHN, Split CIFAR100, Split miniIMN), 32 (CORe50)
  - – Barlow Twins's trade-off term $\lambda_{BT}$: 2e-3 (all benchmarks)
  - – Barlow Twins's moving average term: 0.3 (all benchmarks)
  - – Fast learner's trade-off term $\lambda_{train}$: 2.0 (all benchmarks)
  - – Soft label loss temperature $\tau$: 2.0 (all benchmarks)
  - – Number of inner updates: 2 (all benchmarks)
  - – Number of outer updates: 2 (all benchmarks)

- MuFAN
  - – SGD Learning rate: 0.03 (all benchmarks)
  - – Replay batch size: 64 (Split SVHN, Split CIFAR100, Split miniIMN), 32 (CORe50)
  - – Temperature $\tau$ of $\mathcal{L}_{\text{D-CTN}}$: 2 (all benchmarks)
  - – Point-wise distillation balancing factor $\lambda_{\text{D-CTN}}$: 10 (all benchmarks)

- Structure-wise distillation balancing factor $\lambda_{\text{D-CSD}}$: 0.01 (all benchmarks)
- Teacher and student temperature $\tau$ of $\mathcal{L}_{\text{D-CSD}}$: 0.0001, 2 (all benchmarks)
- Number of samples per task to build the relation across tasks $N$: 10 (all benchmarks)
- Number of gradient updates: 2 (all benchmarks)

To test the sensitivity of the proposed total loss to balancing factors, $\lambda_{\text{D-CTN}}$ and $\lambda_{\text{D-CSD}}$, we observed the results with varying values. Table 14 shows that the proposed loss is not largely sensitivity to those factors. In all cases, MuFAN still outperforms the recent state-of-the-art, DualNet. We chose 10 for $\lambda_{\text{D-CTN}}$ and 0.01 for $\lambda_{\text{D-CSD}}$, which showed good performance on average.

Table 14: Sensitivity to balancing factors, $\lambda_{\text{D-CTN}}$ and $\lambda_{\text{D-CSD}}$, in the proposed loss. Bold fonts represent the best performance.

| | $\lambda_{\text{D-CSD}} = 0.01$ | | | $\lambda_{\text{D-CTN}} = 10.0$ | | |
| | $\lambda_{\text{D-CTN}}$ | | | $\lambda_{\text{D-CSD}}$ | | |
| ACC | 1.0 | 10.0 | 100.0 | 0.001 | 0.01 | 0.1 |
|---|---|---|---|---|---|---|
| Cifar100 | 75.26±0.97 | **75.86±0.35** | 74.95±0.68 | 75.77±0.33 | 75.86±0.35 | **75.93±0.38** |
| miniIMN | 74.48±0.90 | **75.40±0.44** | 74.54±1.01 | 75.11±0.87 | 75.40±0.44 | **76.01±1.23** |
| CORe50 | 65.66±2.44 | **67.30±1.57** | 64.70±2.08 | **67.42±2.20** | 67.30±1.57 | 66.38±2.18 |

## F  Online Task-free Continual Learning Setting

To test the scalability of the proposed structure-wise distillation loss in an online task-free CL scenario where the notion of tasks and task boundaries are unavailable during training and inference, we revised the original Eq. 2 as follows:

$$\mathcal{L}_{\text{D-CSD-TF}} = \sum_{j=2}^{u(\mathcal{Y}_{\mathcal{M}})//S} \sum_{i=1}^{N_S * S} l(\psi(m_a(e(\mathcal{X}_{j-1,j}^{i,N}))), \psi(m(e(\mathcal{X}_{j-1,j}^{i,N})))) \in \mathcal{M}, \tag{11}$$

where $u(\cdot)$ counts the number of a unique class, $\mathcal{Y}_{\mathcal{M}}$ denotes all currently stored labels in $\mathcal{M}$. $S$ denotes the criterion of a new potential task and $//$ denotes a floor division. That is, with $S = 10$, the summation of $j$ starts when the number of a unique class in $\mathcal{M}$ is greater than or equal to 20. $N_S$ denotes the number of samples per class. Also, subscript $a$ denotes $\omega \neq u(\mathcal{Y}_{\mathcal{M}})$ where $\omega$ represents a previously saved $u(\mathcal{Y}_{\mathcal{M}})$. That is, we store the softmax output of the samples in $\mathcal{M}$ using the model $m$ when the number of a counted unique class is increased. In addition, the replay buffer is implemented as a reservoir buffer instead of a ring buffer, and all training techniques requiring task ID during training and inference were not used.

## G  Various Pre-trained Models

In Table 15, we ablated over varying pre-trained models as the encoder $e$.

Table 15: Ablation study on various pre-trained models as the encoder $e$ (SSL: Self-supervised learning, and OD: Object detection). Bold fonts represent the best performance.

| ACC ↑ | SVHN | CIFAR100 | CORe50 |
|---|---|---|---|
| EfficientNet-lite0 | **94.8±0.7** | **75.9±0.4** | 67.3±1.6 |
| ResNet18 | 93.9±1.0 | 71.5±0.7 | **67.9±1.5** |
| ResNet50 (SSL) | 91.6±1.3 | 72.5±0.7 | 63.6±1.5 |
| SSDlite (OD) | 94.5±0.8 | 74.6±0.6 | 66.0±1.3 |
| DualNet | 93.9±0.5 | 72.6±0.8 | 57.6±1.4 |

Motivated by Sauer et al. (2021), we investigated an EfficientNet-lite0 and ResNet18 trained by the ImageNet dataset (Deng et al., 2009) in a supervised manner, ResNet50 trained by the ImageNet dataset in a self-supervised manner, and SSDlite trained by the MS COCO object detection dataset.

We confirmed that in most cases, any pre-trained model outperformed the recent state-of-the-art method. For MuFAN, we integrate an EfficientNet-lite0, which consistently showed the best or near-best performance, except for the Split miniIMN benchmark.

# H  COMPUTATION AND MEMORY COST

To further compare the model complexity between MuFAN and the main comparison methods, we counted a FLOP per iteration and memory cost at the final task on the CORe50 benchmark. As shown in Table 16, the difference in memory cost among the three methods is negligible. FLOPs per iteration of DualNet and MuFAN are 349.1B and 392.2B, respectively. Indeed, there is a 43.1B increase in MuFAN compared to DualNet, however, in ACC, MuFAN also exhibits a large increase of 9.66% over DualNet.

Table 16: Comparison on FLOP per iteration and memory cost.

| CORe50 | FLOP per iteration | Memory cost | ACC |
|---|---|---|---|
| ER-Ring | 144.3B | 3419.91 MB | 45.11 |
| DualNet | 349.1B | 3453.26 MB | 57.64 |
| MuFAN (Ours) | 392.2B | 3501.99 MB | 67.30 |

# I  T-SNE

In Figure 4, we observed that the relation between cross-task data points is well maintained throughout the stream of tasks.

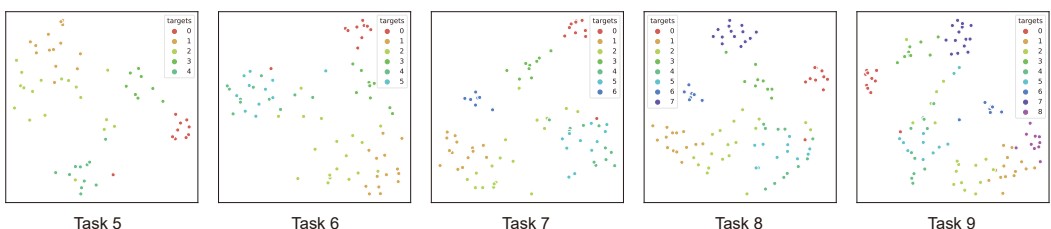

Figure 4: Visualization of the latent space of MuFAN using T-SNE on CORe50. For better visualization, we used data points with a single class from each task.

# J  DIFFERENT DISTANCE METRICS

In addition to CS, we tested other distance metrics to check where there is a better way to penalize the distance. First, we tested with a 2-norm distance as follows:
$$\mathrm{d}(u, v) = \|u - v\|_2 ,\tag{12}$$
where $u$ and $v$ are vectors. Using this distance metric, we revised the original Eq. 3 as follows:
$$\psi(m_{t-1}^*(e(\mathcal{X}_{j-1,j}^{i,N}))) = \sigma(\left\|m_{t-1}^*(h_{j-1}^i) - m_{t-1}^*(h_j^1)\right\|_2, \cdots, \left\|m_{t-1}^*(h_{j-1}^i), m_{t-1}^*(h_j^N)\right\|_2).\tag{13}$$

Second, we tried a different angular distance to check whether there is a better way to penalize the angular distance. In Cer et al. (2018), they proposed a similarity metric that transforms CS as follows:
$$\mathrm{sim}(u, v) = 1 - \arccos(<u, v>)/\pi,\tag{14}$$
where $u$ and $v$ are vectors and $< \cdot >$ denotes CS. This metric computes the CS of two vectors and then uses arccos to create similarity based on an angular distance. Using this similarity metric, we revised the original Eq. 3 as follows:
$$\begin{aligned}\psi(m_{t-1}^*(e(\mathcal{X}_{j-1,j}^{i,N}))) =&\sigma((1 - \arccos(<m_{t-1}^*(h_{j-1}^i), m_{t-1}^*(h_j^1)>)/\pi), \\ &\cdots, (1 - \arccos(<m_{t-1}^*(h_{j-1}^i), m_{t-1}^*(h_j^N)>)/\pi)).\end{aligned}\tag{15}$$

## K    ADDITIONAL ABLATION STUDY ON DISTILLATION LOSS

To test the effectiveness of suppressing forgetting when another task is used to build a relation across tasks rather than consecutive tasks, we revised the original Eq. 2 as follows:

$$\mathcal{L}_{\text{D-FSD}} = \sum_{j=2}^{t-1} \sum_{i=1}^{N} l(\psi(m_{t-1}^{*}(\mathcal{X}_{1,j}^{i,N}))), \psi(m_t(\mathcal{X}_{1,j}^{i,N})))) \in \mathcal{M}, \tag{16}$$

$$\mathcal{L}_{\text{D-LSD}} = \sum_{j=2}^{t-2} \sum_{i=1}^{N} l(\psi(m_{t-1}^{*}(\mathcal{X}_{t-1,j}^{i,N}))), \psi(m_t(\mathcal{X}_{t-1,j}^{i,N})))) \in \mathcal{M}, \tag{17}$$

where $\mathcal{L}_{\text{D-FSD}}$ represents a case that the first task is used, and $\mathcal{L}_{\text{D-LSD}}$ represents a case that the most recently learned task is used.

We tested the number of samples per task $N$ that have to be used to efficiently utilize $\mathcal{L}_{\text{D-CSD}}$. All Split CIFAR100, Split miniIMN, and CORe50 benchmarks contain 5 different classes in each task. As shown in Table 17, there was no difference in performance between $N = 10$ and $N = 20$. There was only a slight degradation in performance for $N = 5$. When $N$ is greater than or equal to the number of classes in each task, it is possible to build the reliable relation across tasks for distillation.

Table 17: Ablation study on the number of samples per task, $N$, for $\mathcal{L}_{\text{D-CSD}}$. Bold fonts represent the best performance.

| | $\mathcal{L}_{\text{D-CSD}}$ (CS) | | |
|---|---|---|---|
| ACC ↑ | $N = 5$ | $N = 10$ | $N = 20$ |
| Cifar100 | 75.4±0.5 | 75.9±0.4 | **76.3±0.7** |
| miniIMN | 75.2±0.6 | **75.4±0.4** | 75.3±0.9 |
| CORe50 | 67.2±1.6 | 67.3±1.6 | **67.6±1.3** |

## L    ADDITIONAL RESULTS OF STABILITY-PLASTICITY NORMALIZATION MODULE

We reported the standard deviation results of Table 6 in Table 18 due to the page limit.

Table 18: Standard deviation results of Table 6 on two continual learning benchmarks considered.

| Norm. | BN | | | RBN | | | IN | | | GN | | | LN | | | SN | | | CN | | | SPN (Ours) | | |
|---|---|---|---|---|---|---|---|---|---|---|---|---|---|---|---|---|---|---|---|---|---|---|---|---|
| STD | ACC | FM | LA | ACC | FM | LA | ACC | FM | LA | ACC | FM | LA | ACC | FM | LA | ACC | FM | LA | ACC | FM | LA | ACC | FM | LA |
| miniIMN | 1.9 | 1.9 | 3.6 | 0.8 | 0.8 | 1.7 | 2.5 | 1.7 | 1.5 | 1.6 | 1.9 | 1.5 | 2.8 | 1.3 | 4.2 | 0.3 | 0.7 | 0.9 | 3.0 | 1.2 | 2.4 | 1.1 | 0.8 | 0.9 |
| CORe50 | 0.5 | 0.5 | 1.1 | 0.7 | 0.7 | 0.6 | 2.6 | 1.2 | 2.8 | 1.3 | 1.0 | 0.8 | 2.2 | 0.9 | 3.3 | 0.7 | 0.6 | 0.6 | 0.9 | 1.0 | 0.4 | 0.9 | 0.9 | 0.2 |

To demonstrate the effectiveness of SPN on existing CL methods, we additionally tested the difference in ER-Ring with SPN and other normalization layers. Table 19 shows that SPN still is superior to both single and multiple normalization layers in ACC and LA. Also, the spatial normalization layer, IN, shows the highest stability. As an ablation study, we assessed SPN using only IN or LN for the stability normalization operation. Huang et al. (2020) and Luo et al. (2018) validated that spatial normalization layers are vulnerable in stable training because they do not utilize a global context. However, because spatial normalization layers do not utilize a global moment, they are more robust to catastrophic forgetting. As shown in the results of Table 20, the combination of two spatial normalization layers can adapt to the characteristic of data in a way of enhancing stable training while maintaining high stability.

Table 19: Effectiveness of SPN on ER-Ring. Bold fonts represent the best performance in each evaluation metric.

| ER-Ring | CORe50 | | |
|---|---|---|---|
| Norm. Layer(s) | ACC ↑ | FM ↓ | LA ↑ |
| BN | 45.11±2.15 | 8.82±0.52 | 50.73±1.81 |
| RBN | 50.76±0.86 | 8.34±1.04 | 56.02±0.98 |
| IN | 44.56±2.66 | **4.84±2.59** | 46.00±1.00 |
| GN | 40.84±1.79 | 7.96±1.23 | 47.80±1.28 |
| LN | 39.54±2.50 | 9.08±1.39 | 45.32±2.18 |
| SN | 50.20±0.39 | 12.08±1.17 | 60.30±0.71 |
| CN | 51.68±0.90 | 9.80±1.74 | 59.28±0.63 |
| BN + IN | 50.38±1.95 | 9.95±1.99 | 60.06±1.51 |
| BN + LN | 48.04±0.87 | 10.20±1.05 | 58.16±1.46 |
| SPN | **52.80±2.34** | 10.64±2.36 | **61.90±1.46** |

Table 20: Ablation study of SPN using only IN or LN for the stability normalization operation on DER++. Bold fonts represent the best performance in each evaluation metric.

| DER++ | miniIMN | | | CORe50 | | |
|---|---|---|---|---|---|---|
| Norm. | ACC | FM | LA | ACC | FM | LA |
| BN + IN | 58.2±0.7 | 9.2±0.6 | 65.4±0.3 | 43.4±1.7 | **7.0±1.9** | 47.5±2.0 |
| BN + LN | 62.7±1.1 | **7.2±0.7** | 67.4±1.4 | 47.5±4.0 | 7.6±3.4 | 52.2±2.9 |
| SPN | **64.8±1.1** | 9.3±0.8 | **72.7±0.9** | **55.0±0.9** | 9.0±0.9 | **62.1±0.2** |

