# OpenReview forum: "New Insights for the Stability-Plasticity Dilemma in Online Continual Learning"
_ICLR.cc/2023/Conference — ICLR 2023 poster_

### Official Review · Reviewer_zSUJ · 2022-10-24

**Confidence:** 5
**Correctness:** 3
**Technical Novelty And Significance:** 2
**Empirical Novelty And Significance:** 2
**Recommendation:** 5

**Clarity, Quality, Novelty And Reproducibility:**

The paper writing needs improvement. Please refer to the strengths and weaknesses section above for details.

**Strength And Weaknesses:**

**Strengths**

1. The paper studies an important problem of online continual learning.
2. The use of a pre-trained model as an encoder and fusing features of the encoder from different scales is a nice idea which is not explored in the context of continual learning extensively.

**Weaknesses**

1. **Overall the method seems very ad-hoc**: While the use of a pre-trained model as an encoder that generates a fused feature map for the classifier is a nice idea, the other two contributions of the paper seem very ad-hoc. The structured-distillation loss seems like a cumbersome way of enforcing knowledge-distillation, and it is not clear if the so-called point-wise knowledge-distillation on more points would achieve similar or better results than structured distillation. The overall training objective (Eq. 4) is the combination of everything, experience-replay, point-wise knowledge distillation, and structured distillation and it is not clear if everything is needed in that objective. Second, the SP-normalization seems even more ad-hoc where half of the features are passed through BatchNorm layers, and the other half are sent to a combination of GroupNorm and InstanceNorm, without any criterion. If BatchNorm is bad for continual learning then there would be excessive forgetting through the half that was sent to the batchnorm. As for Group or InstanceNorm, if the features drift from one task to the next, how well these normalization techniques would be able to cope with forgetting. Overall, it seems that in designing SP-normalization the whole kitchen sync is thrown at the features.

2. **Writing is not clear**: The writing of the paper is not clear and makes the paper a difficult read. The tuple notation for denoting data samples is incredibly cumbersome and non-intuitive. Sec 3.2 is not clear at all, terms are used without any definitions. For example, the functional form of the so-called structure-wise potential $\psi$ is not provided until the end of the section making it difficult to understand what is happening in Eq. 2. Similarly, throughout Sec 3, intuitions are lacking and it seems that everything is combined in an ad-hoc way. What the authors refer to as knowledge-distillation based on ER, is simple experience replay, there is no knowledge-distillation there.


3. **Experiments**:

* Do the baselines, GEM, ER, MER etc also use the pre-trained model? This is not clear from the experiments.

* Do all the baselines use the same codebase?

* Could the authors compare the compute and memory cost of their method vs the baselines?


**Summary Of The Paper:**

The paper proposes a method MuFAN that strikes a better stability-plasticity tradeoff compared to existing methods in an online (single-pass) continual learning setting. The main idea of the method is to use a pre-trained model as an encoder, aggregate the feature maps from different layers, and then pass the aggregated/ fused feature map to a classifier (typically a ResNet model). During training the pre-trained encoder is kept fixed and the classifier is updated. The authors propose a structured knowledge-distillation loss that maintains the angular distance between the successive tasks. Additionally, the authors propose a new normalization technique that splits the channels at a given layer and apply a combination of BatchNorm, InstaceNorm and GroupNorm on different splits of the feature map. The experiments are reported on several supervised continual learning benchmarks.


**Summary Of The Review:**

Please refer to the strengths and weaknesses section above for details.

---

> ### Author Response · Authors · 2022-11-17
> **Response to Reviewer zSUJ (1/2)**
>
> We thank you for the comments and suggestions. We have addressed each of your questions below.
>
> **Q1.** While the use of a pre-trained model as an encoder that generates a fused feature map for the classifier is a nice idea, the other two contributions of the paper seem very ad-hoc.
>
> **Response:** We propose a novel structure-wise distillation loss for CL. This loss should not be considered a simple ad-hoc because there is no such existing distillation loss that enforces the angular similarity of task embedding spaces between consecutive tasks to be maintained. Also, this is the first work that suggests the idea of setting one normalization layer efficient for stability and the other normalization layer efficient for plasticity in parallel to simultaneously address plasticity and stability in online CL. Also, qualitatively, the performance gains by all three components are large. As an example, Table 9 rows (1), (3), and (4) in the main paper on CORe50 showed that when the improvement in ACC through the multi-scale feature map (MF) is 7.29%, the improvements in ACC through the proposed stability-plasticity normalization (SPN) and the structure-wise distillation loss($ \mathcal{L}_{\text{D-CSD}} $) are also 7.32% and 3.17%, respectively. The best performance can be achieved when all three contributions are used together.
>
> We believe that the structural design and loss function of MuFAN are novel as an online CL method.
>
> **Q2.** The overall training objective (Eq. 4) is the combination of everything, experience-replay, point-wise knowledge distillation, and structured distillation and it is not clear if everything is needed in that objective.
>
> **Response:** Table R9 empirically demonstrates the importance of each component. As each component is added, FM continuously decreases. Eventually, the highest ACC and lowest FM can be obtained when the combination of all three components is used.
>
> **<Table R9>**
> | CORe50                                |  ACC &uarr; |  FM &darr;  |
> |---------------------------------------|:-----:|:----:|
> | experience-replay                       | 45.11 | 8.82 |
> | experience-replay + point-wise           | 48.71 | 7.12 |
> | experience-replay +  point-wise + structured | 51.88 | 4.60 |
>
> **Q3.** Second, the SP-normalization seems even more ad-hoc where half of the features are passed through BatchNorm layers, and the other half are sent to a combination of GroupNorm and InstanceNorm, without any criterion. If BatchNorm is bad for continual learning then there would be excessive forgetting through the half that was sent to the batchnorm. As for Group or InstanceNorm, if the features drift from one task to the next, how well these normalization techniques would be able to cope with forgetting. Overall, it seems that in designing SP-normalization the whole kitchen sync is thrown at the features.
>
> **Response:** Batch normalization (BN) is beneficial in improving model plasticity because BN helps quickly adapt to a given task using running statistics within a batch. Therefore, rather than not using BN, we make it robust to data distribution shift by also creating a feature map independent of the statistical inconsistency between training and inference. Just mixing the various normalization layers using learnable weights would be more like a kitchen sync, where considerations on the role of each layer are neglected. However, the motivation of SPN is to allow the convolutional kernels to independently utilize differently normalized feature maps following a given training signal.
>
> **Q4.** The writing of the paper is not clear and makes the paper a difficult read. The tuple notation for denoting data samples is incredibly cumbersome and non-intuitive. Sec 3.2 is not clear at all, terms are used without any definitions.
>
> **Response:** Sorry for the complexity. To make the notation and equation easier to understand, the notation and equation have been revised as follows:
>
> $$
> \mathcal{L}_{\text{D-CSD}} = \sum^{t-1}\_{j = 2} \sum^{N}\_{i = 1}  l (\psi\_{m^*\_{t-1}}(z^i\_{j-1}, \mathcal{Z}^N\_j), \psi\_{m\_t}(z^i\_{j-1}, \mathcal{Z}^N\_j)),
> $$
>
> in a way of not using the tuple notation. Based on your suggestion, we revised the overall equations of Section 3.2 in the main paper.

---

> > ### Comment · Reviewer_zSUJ · 2022-12-04
> > **Rebuttal Acknowledgment**
> >
> > I thank the authors for their response and the changes that they have made to the draft to make it more clear.
> >
> > - **SPN**: I am still not fully convinced by splitting the features in two halves and running different normalization schemes through separate halves. It is not clear to me why normalizing half the features along $[B, H, W]$ dimensions and the other half across $[C, H, W]$ would reduce the covariate shift problem between different layers (which was the main motivation behind the normalization schemes). Even keeping that aside, the authors in their rebuttal claim that BN should improve the plasticity of the network resulting in a better learning of a given task. If this were to be the case then it should be reflected in a better learning accuracy $LA$ of BN (compared to SPN). Table 6 in the paper shows this is not the case. So I am not sure how to interpret the SPN results.
> >
> > - **Using pre-trained models with other baselines**: I thank the authors for running this experiment. However, I am quite surprised that using a pre-trained model did not improve the metrics for something like ER. At the very least, I was expecting the $LA$ to go significantly up on CIFAR100 when initialized from an ImageNet pre-trained model. Without sounding too coarse, I think these experiments may not be run in an optimal setting for these baselines so I am not sure what to make of the numbers. Additionally, I feel that the fact that these baselines were not initialized with a pre-trained model made the comparison quite unfair in the first draft. I believe this is something that authors should've clearly mentioned in their original draft.
> >
> > Overall, I feel that the work can be improved by providing better explanations of why each component is working and running a more careful set of experiments. I will retain my original score.

---

> ### Author Response · Authors · 2022-11-17
> **Response to Reviewer zSUJ (2/2)**
>
> **Q5.** Similarly, throughout Sec 3, intuitions are lacking.
>
> **Response:** Based on your comment, we added and revised sentences in Section 3 to provide more intuition about the proposed components. However, due to the page limit, we also included the intuition about each component in Supplementary. Here we summarize these parts that can obtain additional intuition in detail.
>
> For the multi-scale feature map, we summarized the characteristics of each hierarchical module and other details in Supplementary B. For the structure-wise distillation loss, we assessed that the intuition of a novel structure-wise distillation loss indeed holds, we visualized the latent space of MuFAN using T-SNE in Supplementary I. We observed that a relation between cross-task data points is well maintained throughout a stream of tasks. As last, for the stability-plasticity normalization module, we conducted additional ablation studies about spatial normalization and summarized them in Supplementary J. We hope that these experiments and analyses help you understand the intuitions in MuFAN.
>
> **Q6.** What the authors refer to as knowledge-distillation based on ER, is simple experience replay, there is no knowledge-distillation there.
>
> **Response:** Based on your comment, we modified $ \mathcal{L}\_{\text{D-ER}} $ to $ \mathcal{L}\_{\text{ER}} $  and revised the writing of Section 3.2 in the main paper accordingly.
>
> **Q7.** Do the baselines, GEM, ER, MIR etc also use the pre-trained model? This is not clear from the experiments. Do all the baselines use the same codebase?
>
> **Response:**  Yes, we use the same codebase on all baselines and MuFAN. Because our contribution to the use of the pre-trained model is using it as an encoder, MuFAN and all baselines start from the randomly initialized backbone. However, based on your suggestion, we tested the performance of GEM, ER-Ring, and MIR when an ImageNet pre-trained ResNet18 is used as a backbone rather than a randomly initialized ResNet18. As shown in Table R10, there was no change in performance on Split CIFAR100, which is a collection of natural images similar to ImageNet. However, Split SVHN, which is a collection of street number images, showed a performance drop of 4.12%, 6.82%, and 7.99% in ACC on GEM, ER-Ring, and MIR, respectively, when using an ImageNet pre-trained backbone. It seems that an ImageNet pre-trained model is rather biased on natural images, so it does not become a beneficial starting point even compared to the randomly initialized model.
>
> **<Table R10>**
> |                                   |       | Split SVHN |       |       | Split CIFAR100 |       |
> |-----------------------------------|:-----:|:----------:|:-----:|:-----:|:--------------:|:-----:|
> |                                   |  ACC &uarr;  |     FM &darr;     |   LA &uarr;  |  ACC &uarr;  |       FM &darr;       |   LA &uarr;  |
> | GEM                               |  82.30 |    12.16   | 91.06 | 57.89 |      8.62      | 63.01 |
> | GEM (w/ Pre-trained Backbone)     |  78.18 |    13.16   | 85.06 | 59.76 |      9.77      | 66.72 |
> | ER-Ring                           | 91.68 |    5.26    | 95.48 | 61.32 |      5.16      | 63.20 |
> | ER-Ring (w/ Pre-trained Backbone) | 84.86 |    5.52    | 83.64 | 61.45 |      6.59      | 65.44 |
> | MIR                               | 91.22 |    6.18    | 96.16 | 64.97 |      7.78      | 70.03 |
> | MIR (w/ Pre-trained Backbone)     | 83.23 |    6.02    | 82.11 | 63.66 |      7.99      | 71.67 |
>
>
> **Q8.** Could the authors compare the compute and memory cost of their method vs the baselines?
>
> **Response:** Based on your suggestion, we counted a FLOP per iteration and memory cost at the final task on the CORe50 benchmark. As shown in Table R11, the difference in memory cost among the three methods was negligible. FLOPs per iteration of DualNet, a recent state-of-the-art online CL method, and MuFAN were 349.1B and 392.2B, respectively. Indeed, there was a 43.1B increase in MuFAN compared to DualNet, however, in ACC, MuFAN also exhibited a large increase of 9.66% over DualNet.
>
> **<Table R11>**
> | CORe50       | FLOP per iteration | Memory cost |  ACC &uarr;  |
> |--------------|:------------------:|:-----------:|:-----:|
> | ER-Ring      | 144.3B |   3419.91 MB   | 45.11 |
> | DualNet      | 349.1B |   3453.26 MB   | 57.64 |
> | MuFAN (Ours) | 392.2B |   3501.99 MB   |  67.30 |

---

### Official Review · Reviewer_L1XG · 2022-10-24

**Confidence:** 4
**Correctness:** 3
**Technical Novelty And Significance:** 3
**Empirical Novelty And Significance:** 3
**Recommendation:** 8

**Clarity, Quality, Novelty And Reproducibility:**

Quality: The authors have shown significant improvement in online continual learning especially when there is a task-free setting in addition to outperforming most SOTA methods in a standard setting.

Novelty: Although more recent approaches like [1, 2] have used knowledge distillation between models as a similarity function (some settings of these approaches differ), using a structural distillation function for hierarchical training is a somewhat mildly explored concept, the authors have used this effectively to maintain the similarity of the tasks embedding space with the model as it is learning incrementally in an online fashion. In addition to that, the encoder-based top-down approach in this paper is somewhat explored in this work [3]; however, the authors have shown that multi-scale feature extraction, similar to U-net, can improve the accuracy significantly.

Reproducibility: based on the datasets the authors have used, the detailed methodology, and the clarity of the work, I believe this work is reproducible.

[1] M. H. Phan, T.-A. Ta, S. L. Phung, L. Tran-Thanh, and A. Bouzerdoum, “Class Similarity Weighted Knowledge Distillation for Continual Semantic Segmentation,” in 2022 IEEE/CVF Conference on Computer Vision and Pattern Recognition (CVPR), New Orleans, LA, USA, Jun. 2022, pp. 16845–16854. doi: 10.1109/CVPR52688.2022.01636.

[2] M. Kang, J. Park, and B. Han, “Class-Incremental Learning by Knowledge Distillation with Adaptive Feature Consolidation,” in 2022 IEEE/CVF Conference on Computer Vision and Pattern Recognition (CVPR), New Orleans, LA, USA, Jun. 2022, pp. 16050–16059. doi: 10.1109/CVPR52688.2022.01560.

[3] A. Rannen, R. Aljundi, M. B. Blaschko, and T. Tuytelaars, “Encoder Based Lifelong Learning,” in 2017 IEEE International Conference on Computer Vision (ICCV), Venice, Oct. 2017, pp. 1329–1337. doi: 10.1109/ICCV.2017.148.


**Strength And Weaknesses:**

Strengths
1. The paper essentially describes a novel way to address the problem of catastrophic forgetting.
   a. First, similar to the architectures used in the domain of semantic segmentation, extracting the features from shallow to deep layers becomes essential especially when extracting deep high-level features and shallow low-level features. In this task, the extraction of multi-scale feature maps helps the model learn its intrinsic features when implying the addition of the top-down module.
   b. Although the significance of sequential hierarchical training in the domain of continual learning has not been explored widely, the authors have used a similar concept to force the model to learn the relationship between two consecutive tasks by maintaining an angular distance. Based on their ablation study, this has shown some improvements in stability, but not so much in accuracy and plasticity.
   c. The proposed parallel stability-plasticity normalization module tackles the variability issues of the incoming data and problems of normalization in the inference step. For example, the inconsistency in the data during training and inference can cause a significant drop in the performance of the previous task, leading to catastrophic forgetting. Here, this method finds a balance between stability and plasticity.
2. The authors have performed comprehensive experiments against other SOTA methods. Their extensive ablation study further solidifies their claims.
3. The authors have made use of all the most common benchmark datasets that are widely used in this area, furthering the compatibility of their approach in regard to its reproducibility.
4. The paper is easy to follow.

Weaknesses
1. Why are the ablation studies in Tables 3 and 4 performed on ER-Ring? Also, it seems to be the case with table 6 as well with DER++. The authors’ proposed approach is MuFAN. (It is either not clearly stated in the paper or there can be a  typo)
2. In table 9, it can be observed that by using all three proposed components and data augmentation, only the best overall accuracy can be achieved. However, as stated by the authors, for tackling the stability-plasticity dilemma,  row-8 has shown significant improvement in handling stability whereas row-7 handles plasticity quite well. Therefore, what is affecting these components when used altogether?
3. The authors have mentioned that the architecture is similar to that of U-Net and discussed the improvements over utilizing standard and bottom-up extraction methods; however, as mentioned in this work [1], it would be interesting to see the compatibility of the top-down (without multi-scale feature extraction) approach with auto-encoders in their ablation study.

[1] A. Rannen, R. Aljundi, M. B. Blaschko, and T. Tuytelaars, “Encoder Based Lifelong Learning,” in 2017 IEEE International Conference on Computer Vision (ICCV), Venice, Oct. 2017, pp. 1329–1337. doi: 10.1109/ICCV.2017.148.


**Summary Of The Paper:**

The authors address the problem of stability and plasticity in continual learning in a fully online manner. They do that by introducing the extraction of multi-scale feature maps from shallow and deep layers of a pre-trained model, a structure-wise distillation loss across multiple tasks, and a parallel normalization module for stability and plasticity. The proposed approach outperforms most of the current SOTA methods in regard to accuracy (ACC), stability (FM), and plasticity (LA). In addition, the authors have performed extensive experiments in their ablation study foolproofing the significance of their approach.

**Summary Of The Review:**

The authors have shown improvements over SOTA methods in the online continual learning paradigm. Their claims are supported with empirical evidence. In addition, each of the components in their work are contributing to the approach. I believe this would be a good contribution. Therefore, I am leaning towards accepting this paper; however, the authors will need to clarify the issues mentioned in the weakness section.

---

> ### Author Response · Authors · 2022-11-17
> **Response to Reviewer L1XG**
>
> We thank you for the positive comments and insightful suggestions. We have addressed each of your questions below.
>
> **Q1.** Why are the ablation studies in Tables 3 and 4 performed on ER-Ring? Also, it seems to be the case with table 6 as well with DER++. The authors’ proposed approach is MuFAN.
>
> **Response:** To check the effectiveness of the proposed components more clearly, we first tested each component with the baseline CL method (ER and DER++) in several experiments (Tables 3, 4, and 6). Based on these evaluations, in Table 9 in the main paper, we tested the effectiveness of each proposed component of MuFAN by ablation studies. Sorry for the confusion. We added a sentence explaining these experiments in Section 4.2 of the main paper.
>
> **Q2.** In table 9, it can be observed that by using all three proposed components and data augmentation, only the best overall accuracy can be achieved. However, as stated by the authors, for tackling the stability-plasticity dilemma, row-8 has shown significant improvement in handling stability whereas row-7 handles plasticity quite well. Therefore, what is affecting these components when used altogether?
>
> **Response:**
> When the multi-scale feature map obtained from the encoder is fed into the backbone (Table 9 row (1)), LA largely increases, but FM also largely increases while learning a sequence of tasks. On top of that, we distill the relationship between previously learned tasks to give constraints to the change of knowledge. This provokes less forgetting and less plasticity at the same time (Table 9 row (8)). As last, the commonly used batch normalization brings additional forgetting in CL due to training and inference statistics mismatching. To alleviate this problem while maintaining high learning performance for new tasks, we utilize spatial normalization in a parallel way. That is, when the distillation loss sacrifices LA for FM, the SPN helps raise LA while maintaining FM. That is why Table 9 row (11) in the main paper showcases the best ACC.
>
> **Q3.** The authors have mentioned that the architecture is similar to that of U-Net and discussed the improvements over utilizing standard and bottom-up extraction methods; however, as mentioned in this work [1], it would be interesting to see the compatibility of the top-down (without multi-scale feature extraction) approach with auto-encoders in their ablation study.
>
> **Response:** Based on your suggestion, without multi-scale feature extraction, we checked the performance of the approach with auto-encoders [1]. In experiments, we passed the last convolutional feature map from the pre-trained encoder through a learnable decoder. The decoder consists of layers of 2D transposed convolution with batch normalization and ReLU. As shown in Table R8 row (3), the result of the approach with auto-encoders was even lower than the standard and bottom-up methods. It seems that the feature map from the decoder is not enough to train a full backbone.
>
> **<Table R8>**
> | ACC &uarr;                  | SVHN | CIFAR100 |
> |---------------------|:----:|:--------:|
> | (1) ER-Ring (Fixed, ST) | 63.6 |   61.9   |
> | (2) ER-Ring (Fixed, BU) | 65.8 |   64.0   |
> | (3) ER-Ring (Auto-encoder)   | 60.9 |   47.3   |
> | (4) ER-Ring (TD)        | 92.3 |   69.3   |

---

> > ### Comment · Reviewer_L1XG · 2022-11-28
> > **Acknowledgement**
> >
> > Thank you for the detailed response. It clarifies the major points.

---

### Official Review · Reviewer_dVei · 2022-10-24

**Confidence:** 4
**Correctness:** 3
**Technical Novelty And Significance:** 1
**Empirical Novelty And Significance:** 3
**Recommendation:** 5

**Clarity, Quality, Novelty And Reproducibility:**

The paper is very dense. The authors do a reasonable job explaining most of the contributions, however some parts are hard to read : for example, please consider using a different notation than $\mathcal{X}^{1,2}_{i,j}$ : you could just list the subsets in task $i$ and task $j$ one after the other.



**Strength And Weaknesses:**

Strengths :
The paper positions itself well in the current literature.
The paper is clear in detailing has many ablations to evaluate the utility of the proposed components

Weaknesses :
1. *On the stability / plasticity conclusions drawn from Figure 1*.
The authors argue because online methods have a lower accuracy after a training task (LA), then they are less plastic. I strongly believe that this is property is not intrinsic to online learning. Rather, this property is a direct result of **the number of training iterations spent on each task.** An easy way to increase the number of training iterations in the online setting is to simply do multiple gradient optimization steps (each with different rehearsal batches) as in ([1], [2]). It would be good for the authors to redo figure 1, where the number of iterations on a datapoint is equal to the number of epochs in the offline setting (so that online and offline methods use the same compute).
2. *On the use of pretrained models in online CL* .
The authors propose a novel (to the best of my knowledge) yet quite convoluted way to leverage pretrained models. Yet, highly simple and straightforward baselines (such as a nearest-class-mean classifier in the feature space of the pretrained model, or downright finetuning of such model) are omitted. Given that there is ample evidence [3,4,5] that these methods are surprisingly effective, it is crucial that one compares to such naive approaches : if one cannot beat them, then why bother with the complexity of a new method ?
Here is a small code snippet [6] that evaluates one such simple approach using the setting described on the Split CIFAR100 setting. **Without any training, one can boost the accuracy from 75% to 85% using solely the pretrained model!**
3. *On the added complexity of the method*
From the paper, it is unclear how many additional forward passes are required at each step. Looking at 1 and 2, it seems that the full buffer is forwarded through the model at each step ? If this is the case, it is important to note that this comes at a very high additional cost in compute, and should be properly discussed. Also, what is the cost of the additional forward passes through the old model for distillation ? Given that data augmentations are used, I am assuming hidden activations cannot be cached and have to be recomputed ?
On a similar note, it would be much more interesting to replace table 8 with a FLOP per iteration count for several methods.
Similarly, what is the additional compute cost of using a top-down module ? my understanding is that upsampling the representation before feeding it to the classifier will have a significant increase in compute cost. It would be nice if the authors could analyse this.
4. *On the task-free version of MuFAN*
This version uses a hyperparameter $\mathcal{S}$, which essentially determines the size of each task (in number of unique labels). In a "real" online setting, the learner does not know how long the learning stream is, nor does it know how many unique labels it will see. Therefore, how would one set $\mathcal{S}$ a priori ? Moreover, in this section you are comparing to baselines which **do not** use a pretrained model; I don't see how this is a fair comparison



[1] Drinking from a Firehose: Continual Learning with Web-scale Natural Language https://arxiv.org/abs/2007.09335

[2] Online Continual Learning with Maximally Interfered Retrieval https://arxiv.org/abs/1908.04742

[3] REMIND Your Neural Network to Prevent Catastrophic Forgetting https://arxiv.org/abs/1910.02509

[4] A Simple Baseline that Questions the Use of Pretrained-Models in Continual Learning https://arxiv.org/abs/2210.04428

[5] Lifelong Machine Learning with Deep Streaming Linear Discriminant Analysis https://arxiv.org/abs/1909.01520
[6]
```
import os
import PIL
import torch
from torchvision import datasets, transforms

if not os.path.exists('efficientnet_lite0.pth'):
    os.system('wget https://github.com/RangiLyu/EfficientNet-Lite/releases/download/v1.0/efficientnet_lite0.pth')

if not os.path.exists('eff_net_code'):
    os.system('git clone git@github.com:RangiLyu/EfficientNet-Lite.git eff_net_code')

# load torch model
from  eff_net_code.efficientnet_lite import build_efficientnet_lite
model = build_efficientnet_lite('efficientnet_lite0', 1000)
model.load_state_dict(torch.load('efficientnet_lite0.pth'))
model = model.cuda()
print(f'total of {sum(x.numel() for x in model.parameters()) / 1e6 :.2f} M params.')

# remove linear head
model.dropout = model.fc = torch.nn.Identity()
model.avgpool = torch.nn.Identity()
DIM = model(torch.cuda.FloatTensor(2, 3, 224, 224)).size(-1)

# default transforms for the model in question
tfs = transforms.Compose([
    transforms.Resize(224 + 32, interpolation=PIL.Image.BICUBIC),
    transforms.CenterCrop(224),
    transforms.ToTensor(),
    transforms.Normalize([0.498, 0.498, 0.498], [0.502, 0.502, 0.502])
    ])

ds_train = datasets.CIFAR100('./', train=True, download=True, transform=tfs)
ds_test  = datasets.CIFAR100('./', train=False, download=True, transform=tfs)
BS, n_cls, n_way = 256, 100, 5 # use n_way == 100 for the single-head task-free result


@torch.no_grad()
def get_prototypes(model, dataset, n_classes):
    protos = torch.zeros(n_classes, DIM).cuda()
    counts = torch.zeros(n_classes, dtype=torch.int64).cuda()

    loader = torch.utils.data.DataLoader(dataset, batch_size=BS, num_workers=8, shuffle=True)
    offset = torch.arange(DIM).cuda()

    print(f'processing : {(str(type(dataset)))}')
    for i, (x,y) in enumerate(loader):
        print(f'{i}/{len(loader)}', end='\r')
        x, y = x.cuda(), y.cuda()
        feats = model(x)

        # accumulate
        counts.scatter_add_(0, y, torch.ones_like(y))
        idx = offset.view(1, -1) + y.view(-1, 1) * DIM
        protos.view(-1).scatter_add_(0, idx.view(-1), feats.view(-1)).view_as(protos)

    return protos / counts.unsqueeze(-1)


@torch.no_grad()
def evaluate(model, dataset, protos, n_way):
    loader = torch.utils.data.DataLoader(dataset, batch_size=BS, num_workers=8, shuffle=True)
    n_cls = protos.size(0)

    n_ok, n_tot = 0, 0

    for i, (x,y) in enumerate(loader):
        print(f'{i}/{len(loader)}', end='\r')
        x, y = x.cuda(), y.cuda()
        feats = model(x)

        dist = (feats.unsqueeze(1) - protos.unsqueeze(0)).pow(2).mean(-1)
        pred = dist.argmin(1)
        acc  = (pred == y).float().mean()

        # since in CL we typically partition and eval on a subset of classes, let's mimick this
        _, preds_in_order = dist.sort(1, descending=False)
        pos_of_correct_answer = torch.where(preds_in_order == y.unsqueeze(-1))[1]

        n_extra = n_way - 1

        # if we pick any `n_extra` classes at random, what is p(correct) ?
        p_correct = dist.new_ones(size=y.size())
        for it in range(n_extra):
            n_worse_left = ((n_cls - 1) - (pos_of_correct_answer) - it).clamp_(min=0)
            n_better = pos_of_correct_answer # fixed
            p_correct = p_correct * (n_worse_left / (n_worse_left + n_better))

        n_ok  += p_correct.sum().item()
        n_tot += x.size(0)

    print(f'\nTest Acc : {n_ok / n_tot  * 100 :.2f}')

protos = get_prototypes(model, ds_train, n_cls)
evaluate(model, ds_test, protos, n_way)
```

**Summary Of The Paper:**

The paper argues that a good continual learner should strike a different stability - plasticity tradeoff in online and offline CL, the former being more stable and the latter more plastic. The authors propose a new method MuFAN, which comprises 3 new components : the use of multi-scale features of pretrained models, an added cross-task distillation loss, and a new normalization technique. The authors evaluate their method on standard online CL benchmarks, and provide some ablation studies to highlight the effectiveness of the components.

**Summary Of The Review:**

Overall, given that the main contribution relies on proposing a new way to leverage pretrained models, but that key baselines are missing from this analysis, my opinion is that the paper should make these changes and be resubmitted. A proper evaluation of the computation complexity of the method is missing as well. While these two key issues are not properly addressed, my rating will stay the same.

Post rebuttal : The authors have addressed some of my concerns, so I am raising my score accordingly.

---

> ### Author Response · Authors · 2022-11-17
> **Response to Reviewer dVei (1/3)**
>
> We thank you for the comments and suggestions. We have addressed each of your questions below.
>
> **Q1.** On the stability / plasticity conclusions drawn from Figure 1. The authors argue because online methods have a lower accuracy after a training task (LA), then they are less plastic. I strongly believe that this is property is not intrinsic to online learning. Rather, this property is a direct result of the number of training iterations spent on each task. An easy way to increase the number of training iterations in the online setting is to simply do multiple gradient optimization steps (each with different rehearsal batches) as in ([1], [2]). It would be good for the authors to redo figure 1, where the number of iterations on a datapoint is equal to the number of epochs in the offline setting (so that online and offline methods use the same compute).
>
> **Response:** In the conventional scenario of online learning, they assume data that comes strictly in a streaming manner. That is, the model immediately adapts to data at that moment, and the streaming instance or small batch is immediately discarded [A]. Therefore, storing all of the streaming data in the online validation buffer during a short moment suggested in the above papers ([1], [2]) is not suitable for the conventional online learning scenario. However, based on your suggestion, we tested whether equaling the number of iterations and the number of epochs brings similar results. As shown in Table R2, even if the number of iterations and the number of epochs match one another, we still observed lower LA and lower FM in online CL. In other words, the characteristics of online CL are maintained unless offline storage is assumed.
>
> **<Table R2>**
> |            |       | CIFAR100 |       |       | miniIMN |       |       | CORe50 |       |
> |------------|:-----:|:--------:|:-----:|:-----:|:-------:|:-----:|:-----:|:------:|:-----:|
> |            |  ACC &uarr;  |    FM &darr;   |   LA &uarr;  |  ACC &uarr;  |    FM &darr;   |   LA &uarr;  |  ACC &uarr;  |   FM &darr;   |   LA &uarr;  |
> | 10 Epochs | 60.38 |   17.67  | 77.16 | 59.14 |  20.02  | 78.10 | 53.68 |  11.76 | 65.32 |
> | 10 Iterations    | 58.93 |   13.40  | 71.62 | 58.70 |   9.87  | 67.13 | 51.70 |  9.30  | 59.05 |
>
>
> [A] Polikar, Robi, et al. "Learn++: An incremental learning algorithm for supervised neural networks." IEEE transactions on systems, man, and cybernetics, part C (applications and reviews) 31.4 (2001): 497-508.
>
> **Q2.** On the use of pretrained models in online CL . The authors propose a novel (to the best of my knowledge) yet quite convoluted way to leverage pretrained models. Yet, highly simple and straightforward baselines (such as a nearest-class-mean classifier in the feature space of the pretrained model, or downright finetuning of such model) are omitted. Given that there is ample evidence [3,4,5] that these methods are surprisingly effective, it is crucial that one compares to such naive approaches : if one cannot beat them, then why bother with the complexity of a new method ? Here is a small code snippet [6] that evaluates one such simple approach using the setting described on the Split CIFAR100 setting. Without any training, one can boost the accuracy from 75% to 85% using solely the pretrained model!
>
> **Response:** Based on your suggestion of downright finetuning, we tested the performance of ER-Ring when an ImageNet pre-trained ResNet18 is used as a backbone rather than a randomly initialized ResNet18. As shown in Table R3, there was no change in performance on Split CIFAR100, which is a collection of natural images similar to ImageNet. However, Split SVHN, which is a collection of street number images, showed a performance drop of 6.82% in ACC when using an ImageNet pre-trained backbone. It seems that an ImageNet pre-trained model is rather biased on natural images, so it does not become a beneficial starting point even compared to the randomly initialized model.
>
> **<Table R3>**
> |                                   |       | Split SVHN |       |       | Split CIFAR100 |       |
> |-----------------------------------|:-----:|:----------:|:-----:|:-----:|:--------------:|:-----:|
> |                                   |  ACC &uarr;  |     FM &darr;    |   LA &uarr;  |  ACC &uarr;  |       FM &darr;      |   LA &uarr; |
> | ER-Ring                           | 91.68 |    5.26    | 95.48 | 61.32 |      5.16      |  63.2 |
> | ER-Ring (w/ Pre-trained ResNet18) | 84.86 |    5.52    | 83.64 | 61.45 |      6.59      | 65.44 |
>
> (Continued)

---

> > ### Comment · Reviewer_dVei · 2022-11-18
> > **Response**
> >
> > Thank you for prodiving a detailed response to my initial concerns.
> >
> > *On Figure 1*
> >
> > Thank you for the additional experiment. There seems to be a misunderstanding of what I meant by the number of iterations. Here is what I meant. A regular ER step loops like this
> > ```
> > for timestep t in stream:
> >     x_t, y_t ~ receive_data()
> >     x_re, y_re ~ buffer_sample()
> >     SGD_step on loss( model( (x_t,x_re), (y_t,y_re) ) )
> > ```
> > What I am proposing is this :
> > ```
> > for timestep t in stream:
> >     x_t, y_t ~ receive_data()
> >     for iteration in [1, ..., N_iter]:
> >         x_re, y_re ~ buffer_sample()
> >         SGD_step on loss( model( (x_t,x_re), (y_t,y_re) ) )
> > ```
> > Note that **there is no need to store additional data compared to regular ER** and this does not violate the streaming assumption of Online CL.
> >
> > *On using pretraining models and NCM classifiers*
> >
> > Thank you for the detailed rebuttal. It is interesting to see the link between pretraining and test domains as the authors pointed out. Regarding the SVHN results : from Table 1 in the paper it looks like MUFAN gets `94.8%` accuracy on split SVHN. Since this number matches the number with the efficientNet-lite0 backbone (Table 1 page 6), I will assume that the setting in the rebuttal is the same as the setting in the paper, which says that SVHN is split into 5 tasks, each a **2 way** classification.
> >
> > I honestly don't understand how you find that `NCM Classifier (EfficientNet-lite0)` gets only 57% in this 2-way classification setting. I have provided you with code that implements NCM classifier. If you change the following code to
> > ```
> > ds_train = datasets.SVHN('./', split='train', download=True, transform=tfs)
> > ds_test  = datasets.SVHN('./', split='test', download=True, transform=tfs)
> > BS, n_cls, n_way = 256, 10, 2
> > ```
> > You get **83%** accuracy. I have validated this exact code both on my local machine and on google colab to make sure. I encourage you to do the same.
> > On the 5-way setting, I get  **63%** again on my local machine and on google colab. I don't understand how you get **46%**
> >
> > I realise the the 2-way results are indeed lower than MU-FAN, and I do agree that MU-FAN can offer gains in settings where pretraining and CL domains are different.
> >
> > *On the compute cost*
> >
> > Thank you for the clarification. This does answer my concerns regarding this point.
> >
> >
> >
> > Overall, while the authors have answered some of my points, I do have several remaining concerns. I will increase my score to 4, as the authors have provided a decent rebuttal. While this paper can make for a good submission in the future, provided that authors revamp the experimental section to include proper benchmarking, I don't think the current version meets the ICLR bar.

---

> > > ### Author Response · Authors · 2022-11-19
> > > **Re: Response**
> > >
> > > We would like to thank you for the fast response. We are glad that you leave additional questions about your concern.
> > >
> > > *On Figure 1*
> > >
> > > **Q1.** Note that there is no need to store additional data compared to regular ER and this does not violate the streaming assumption of Online CL.
> > >
> > > **Response.** The results of Table R2 were obtained by running the code just like the code snippet you uploaded. We think that the notion of storage is different between us. We believe that increasing the number of iterations means that the batch should be preserved (stored) in the memory during the iterations. As the Reviewer aHBY mentioned, online learning basically assumes a short training time. Thus, it is not a practical approach to increase the number of iterations significantly in online learning. However, in summary, the results of Table R2 were obtained in exactly what you suggested. And, we still observed the inherent characteristics of the stability-plasticity dilemma in online learning.
> > >
> > > *On using pretraining models and NCM classifiers*
> > >
> > > **Q2.** I honestly don't understand how you find that NCM Classifier (EfficientNet-lite0) gets only 57% in this 2-way classification setting.
> > >
> > > **Response.** We reported the Table R4 rows (1-3) by using the code and libraries uploaded in the official github code of [4]. Please leave a comment if you obtain different results by running it.
> > >
> > > **Q3.** You get 83% accuracy. I have validated this exact code both on my local machine and on google colab to make sure. I encourage you to do the same. On the 5-way setting, I get  63% again on my local machine and on google colab. I don't understand how you get 46%
> > >
> > > **Response.** Sorry for the confusion. We followed your code snippet except for the line loading the pre-trained model. We utilized the pre-trained model of the timm library [A] as shown below
> > >
> > > *model = timm.create_model("efficientnet_lite0", pretrained=True).cuda()*
> > >
> > > instead of the model from the github (https://github.com/RangiLyu/EfficientNet-Lite) (you suggested) for a fair comparison between [4], MuFAN, and your code snippet.
> > >
> > > As you explained, for a fair comparison, we tested the results of a 2-way. We observed an accuracy of 72.77% on SVHN with the pre-trained model provided in timm. Since both the reported results of our paper (MuFAN) and [4] were obtained based on the timm library, we replaced that line with the timm model. We believe that in the recent literature of the computer vision community, the pre-trained model provided by timm is the most generally used pre-trained model. We also believe that it is hard to insist that it is a reasonable method if the performance greatly varies depending on the pre-trained model or library version. Please leave a comment if you do not obtain an accuracy of 72.77% with timm 0.4.9.
> > >
> > > [A] https://github.com/rwightman/pytorch-image-models/
> > >
> > >
> > > We would like to thank you for increasing the score. However, we believe that the contributions of this work meet the ICLR’s bar. We strongly hope that you judge this work more positively. If you leave a question about the remaining concerns, we will answer it.

---

> > > > ### Comment · Reviewer_dVei · 2022-11-19
> > > > **Thank you**
> > > >
> > > > Thank you for the fast response. Given that all the previous work has been done with the checkpoint you provided, I understand now where the discrepancy comes from. I have updated my score.

---

> ### Author Response · Authors · 2022-11-17
> **Response to Reviewer dVei (2/3)**
>
> (Continued)
>
> Also, based on your suggestion of a nearest-class-mean classifier in the feature space of the pre-trained model as a simple baseline, we checked its performance of it on Split SVHN and CIFAR100 using the official code from [4]. As demonstrated in [4] and Table R4 row (1), high ACC can be obtained on Split CIFAR100 when the ImageNet pre-trained Vision Transformer (ViT) is used. However, this trend was not consistent when utilizing different network models, such as EfficientNet-light0 and ResNet18 (as shown in Table R4 rows (2-3)). Moreover, ViT did not showcase a good performance on Split SVHN (street number images), which is distant from natural images (ImageNet). That is, the performance of the method [4] is heavily dependent on a type of pre-trained neural networks and a domain of testing benchmarks, which is very far from what MuFAN is aiming for. As shown in Table R4 rows (4-7), MuFAN showed consistently high performance regardless of various pre-trained models on both Split SVHN and CIFAR100.
>
> **<Table R4>**
> | ACC &uarr;                                 | Split SVHN | Split CIFAR100 |
> |-------------------------------------|:----------:|:--------------:|
> | (1) NCM Classifier (ViT)              |    50.4    |      88.6      |
> | (2) NCM Classifier (EfficientNet-lite0) |    57.0    |      61.1      |
> | (3) NCM Classifier (ResNet18)           |    51.0    |      61.4      |
> | (4) MuFAN (Ours) (EfficientNet-lite0)           |    94.8    |      75.8      |
> | (5) MuFAN (Ours) (ResNet18)                     |    93.9    |      71.5      |
> | (6) MuFAN (Ours) (ResNet50 (SSL))               |    91.6    |      72.5      |
> | (7) MuFAN (Ours) (SSDlite (OD))                 |    94.5    |      74.6      |
>
> As last, we run the code snippet you uploaded on the SVHN benchmark. By leaving all the settings you provided, we obtained a test accuracy of 46.07% (Table R5), which is way lower than the result of CIFAR100 you mentioned. This result also proved that the CL method using the frozen pre-trained model provokes the domain reliability problem that the existing CL did not have.
>
> **<Table R5>**
> | ACC &uarr; (Code snippet) | 5-way |
> |--------------------|:-----:|
> | CIFAR100        | 82.72 |
> | SVHN               | 46.07 |
>
> In summary, because the suggested straightforward and simple baselines heavily rely on the type of pre-trained neural networks and domain of testing benchmarks, these baselines cannot solve the problem that the original pre-trained model cannot solve. However, MuFAN liberates this limitation by utilizing the pre-trained model as a tool (encoder) to obtain diverse levels of both general and prominent features.
>
> **Q3.** On the added complexity of the method. From the paper, it is unclear how many additional forward passes are required at each step. Looking at 1 and 2, it seems that the full buffer is forwarded through the model at each step?
>
> **Response:** Sorry for the confusion. No, only a small batch from the replay buffer passes through the model, not the full buffer.
>
> **Q4.** It would be much more interesting to replace table 8 with a FLOP per iteration count for several methods.
>
> **Response:** Based on your suggestion, we counted a FLOP per iteration at the final task on the CORe50 benchmark. As shown in Table R6, FLOPs per iteration of DualNet, a recent state-of-the-art online CL method, and MuFAN were 349.1B and 392.2B, respectively. There was a 43.1B increase in MuFAN compared to DualNet, however, in ACC, MuFAN also exhibited a large increase of 9.66% over DualNet.
>
> **<Table R6>**
> | CORe50       | FLOP per iteration |  ACC &uarr;   |
> |--------------|:------------------:|:-----:|
> | ER-Ring      |       144.3B       | 45.11 |
> | DualNet      |       349.1B       | 57.64 |
> | MuFAN (Ours) |       392.2B       | 67.30 |
>
> **Q5.** What is the cost of the additional forward passes through the old model for distillation? Given that data augmentations are used, I am assuming hidden activations cannot be cached and have to be recomputed?
>
> **Response:** Yes, we compute the relation across tasks at the end of the task. The additional cost was a 2.5B FLOP per task, which is relatively small compared to the FLOP per iteration shown above.
>
> **Q6.** Similarly, what is the additional compute cost of using a top-down module? my understanding is that upsampling the representation before feeding it to the classifier will have a significant increase in compute cost. It would be nice if the authors could analyse this.
>
> **Response:** When we replace the top-down module with the standard module as in Figure 3 in Supplementary B, the FLOP per iteration changed from 392.2B to 384.5B, which is a 7.7B decrease. Considering the FLOP per iteration, this increase is relatively not large. However, as shown in Table 3 of the main paper, the effectiveness of the top-down module is significant.

---

> ### Author Response · Authors · 2022-11-17
> **Response to Reviewer dVei (3/3)**
>
> **Q7.** On the task-free version of MuFAN. This version uses a hyperparameter $S$, which essentially determines the size of each task (in number of unique labels). In a "real" online setting, the learner does not know how long the learning stream is, nor does it know how many unique labels it will see. Therefore, how would one set $S$ a priori? Moreover, in this section you are comparing to baselines which do not use a pretrained model; I don't see how this is a fair comparison
>
> **Response:**
>
> To show that the performance of the proposed structure-wise distillation loss is not very sensitive to $ S $, we reported the results of MuFAN in a task-free setting with different $ S $s (5 and 10). Table R7 showed that different $ S $s maintain similar performance. That is, we can expect less forgetting through the structure-wise distillation loss with a reasonable $ S $.
>
> For a fair comparison, we also reported the results of DualNet with the multi-scale feature map from the pre-trained encoder in Table R7 row (2). Still, MuFAN maintained superiority over DualNet with the multi-scale feature map.
>
> **<Table R7>**
> | ACC             | CIFAR100 | miniIMN | CORe50 |
> |-----------------|:--------:|:-------:|:------:|
> | (1) DualNet         |   25.5   |   20.9  |  35.6  |
> | (2) DualNet (w/ MF) |   31.2   |   26.0  |  43.7  |
> | (3) MuFAN ($S$ = 5)   |   39.6   |   34.7  |  47.2  |
> | (4) MuFAN ($S$ = 10)  |   38.2   |   33.3  |  48.5  |
>
>
> **Q8.** however some parts are hard to read : for example, please consider using a different notation than $X_{i,j}^{1,2}$: you could just list the subsets in task $i$ and task $j$ one after the other.
>
> **Response:** Sorry for the complexity. To make the notation and equation easier to understand, the notation and equation have been revised as follows:
>
> $$
> \mathcal{L}_{\text{D-CSD}} = \sum^{t-1}\_{j = 2} \sum^{N}\_{i = 1}  l (\psi\_{m^*\_{t-1}}(z^i\_{j-1}, \mathcal{Z}^N\_j), \psi\_{m\_t}(z^i\_{j-1}, \mathcal{Z}^N\_j)),
> $$
>
> in a way of not using $X_{i,j}^{1,2}$. Based on your suggestion, we revised the overall equations of Section 3.2 in the main paper.

---

### Official Review · Reviewer_aHBY · 2022-11-01

**Confidence:** 4
**Correctness:** 3
**Technical Novelty And Significance:** 2
**Empirical Novelty And Significance:** 2
**Recommendation:** 6

**Clarity, Quality, Novelty And Reproducibility:**

The paper does not clearly describe the pretraining procedure; how do they train the pretraining model - what data, how may epochs, what is the pretext task and etc.

**Strength And Weaknesses:**

**Strength**
- S1: Pretraining the model for continual learning brings a noticeable gain in performance.

**Weakness**
- W1: Pretraining using the online stream and complicated normalization scheme incurs a long pipeline for learning. One of the most important feature in continual learning is the short training time (i.e., online learning). This proposal sacrifice the efficiency for the continual learning.
- W2: Most of the gain comes from using multi-scale feature extracted from the pretrained model, which is an existing idea (not applied to CL context yet). Gains by other proposals are marginal (at the expense of larger model than prior arts (Table. 8).
- W3: The `pretraining' is not well suited to continual learning setup as it learns the streamed data by storing them in advance (this part is not very clear in description) for representation learning. As the continual learning setup encourages to learn the data with limited access over the tasks, the notion of pretraining may not be widely applicable in practice.

**Summary Of The Paper:**

The paper proposes to use a pretrained model for its multi-scale feature map for better performance in continual learning along with cross-task structure-wise distillation and a new normalization layer. The proposed method is compared to recent work including DualNet (NeurIPS 2021), CTN (ICLR 2020), MIR (NeurIPS 2019) and etc on SVHN, CIFAR100, miniImageNet and CORe50 dataset. The proposed method improves the performance over the compared methods.

**Summary Of The Review:**

Given that the paper uses seemingly expensive pretraining schemes and the gain is marginal except the gain by the existing idea of using multi-scale feature of pretrained model, the method is not very interesting to be reported in the community. Moreover the description (esp., the pretraining procedure) is not clear enough to judge the value of the work.

---

> ### Author Response · Authors · 2022-11-17
> **Response to Reviewer aHBY (1/2)**
>
> We thank you for the comments and suggestions. We have addressed each of your questions below.
>
>
> **Q1.** Pretraining using the online stream and complicated normalization scheme incurs a long pipeline for learning. One of the most important feature in continual learning is the short training time (i.e., online learning). This proposal sacrifice the efficiency for the continual learning.
>
> **Response:** We agree that the short training time is an important feature in online CL. Thus, we empirically tested how long the training time per iteration of the baselines and MuFAN (ours) at the final task is required. As shown in Table R1, there was no large difference in required training time between DualNet, a recent state-of-the-art online CL method using self-supervised learning, and MuFAN. We checked that the training time of MuFAN is certainly longer than that of ER-Ring. However, from the performance standpoint, MuFAN showed a 22.19% increase over ER-Ring in ACC, and we believe that the required training time of MuFAN is reasonably short.
>
> **<Table R1>**
> | CORe50       | Training time per iteration |  ACC &uarr;  |
> |--------------|:---------------------------:|:-----:|
> | ER-Ring      |            0.045s           | 45.11 |
> | DualNet      |            0.236s           | 57.64 |
> | MuFAN (Ours) |            0.312s           | 67.30 |
>
> **Q2.** Most of the gain comes from using multi-scale feature extracted from the pretrained model, which is an existing idea (not applied to CL context yet). Gains by other proposals are marginal.
>
> **Response:** We believe that the performance gains by other proposals are not marginal. As an example, Table 9 rows (1), (3), and (4) in the main paper on CORe50 showed that **when the improvement in ACC through the multi-scale feature map (MF) is 7.29%, the improvements in ACC through the proposed stability-plasticity normalization (SPN) and the structure-wise distillation loss($ \mathcal{L}_{\text{D-CSD}} $) are 7.32% and 3.17%, respectively**. As shown in these results, the improvements through SPN and $ \mathcal{L}_{\text{D-CSD}} $, respectively, are by no means small. The best performance can be achieved when all three contributions are used together.

---

> ### Author Response · Authors · 2022-11-17
> **Response to Reviewer aHBY (2/2)**
>
> **Q3.** The `pretraining' is not well suited to continual learning setup as it learns the streamed data by storing them in advance (this part is not very clear in description) for representation learning. As the continual learning setup encourages to learn the data with limited access over the tasks, the notion of pretraining may not be widely applicable in practice. The paper does not clearly describe the pretraining procedure; how do they train the pretraining model - what data, how may epochs, what is the pretext task and etc.
>
> **Response:** First, the detail about the pretraining model is discussed in Supplementary D. As explained in Supplementary D, we did not pretrain the model by ourselves, but rather utilized the pre-trained model provided by the timm [A] and vissl [B] libraries, which are commonly used in the recent literature of the computer vision community. Recently, CL methods using the pre-trained model have been published [C], and many interesting methods based on the libraries were also provided in the computer vision [D]  and natural language process [E] communities. Typically, Prompt tuning [F] and Prefix tuning [G] suggested the new notion of using the pre-trained model for fast adaption. To our knowledge, this is the first work considering how to utilize a pre-trained model in a different way for online CL. We suggest a method named MuFAN which utilizes the pre-trained model as an encoder to obtain both general and class-prominent features for model plasticity, rather than limiting scalability by using the pre-trained model as a backbone. We hope that you can agree with the contributions of our paper.
>
>
>
> [A] https://github.com/rwightman/pytorch-image-models/
>
> [B] https://github.com/facebookresearch/vissl
>
> [C] Xue, Mengqi, et al. "Meta-attention for ViT-backed Continual Learning." Proceedings of the IEEE/CVF Conference on Computer Vision and Pattern Recognition. 2022.
>
> [D] Ren, Sucheng, et al. "A Simple Data Mixing Prior for Improving Self-Supervised Learning." Proceedings of the IEEE/CVF Conference on Computer Vision and Pattern Recognition. 2022.
>
> [E] Schucher, Nathan, Siva Reddy, and Harm de Vries. "The Power of Prompt Tuning for Low-Resource Semantic Parsing." Proceedings of the 60th Annual Meeting of the Association for Computational Linguistics (Volume 2: Short Papers). 2022.
>
> [F] Lester, Brian, Rami Al-Rfou, and Noah Constant. "The Power of Scale for Parameter-Efficient Prompt Tuning." Proceedings of the 2021 Conference on Empirical Methods in Natural Language Processing. 2021.
>
> [G] Li, Xiang Lisa, and Percy Liang. "Prefix-Tuning: Optimizing Continuous Prompts for Generation." Proceedings of the 59th Annual Meeting of the Association for Computational Linguistics and the 11th International Joint Conference on Natural Language Processing (Volume 1: Long Papers). 2021.

---

> > ### Comment · Reviewer_aHBY · 2022-11-17
> > **Re: Response to Reviewer aHBY (2/2)**
> >
> > Thank you for the detailed and thoughtful answer. I value the contribution of the paper in the novelty perspective -- that this is the first work of this kind. But I am still concerned about the fact that we need to provide the all data (that is supposed to be given in a stream) to pretrain the model then use it for the 'replayed' samples for the continual learning. Then, this pretrained model is obviously better at less forgetting as it learned the information of the streamed data in advance. I'd like to discuss further on this with the authors.

---

> > > ### Author Response · Authors · 2022-11-18
> > > **Re: Re: Response to Reviewer aHBY (2/2)**
> > >
> > > First, we would like to thank you for the fast response. We are very glad that you agree with MuFAN’s novelty.
> > >
> > > Here is our first response to your current concern. As mentioned in the main paper, **the datasets used for pre-training and the continual learning task are not the same**. The pre-trained models provided by open libraries, such as timm, vissl, and PyTorch are pre-trained on widely used large-scale natural image collections, such as ImageNet and MS COCO. That is, MuFAN does not assume that the data to be given in a stream should be jointly collected in advance. As an example, to evaluate MuFAN on the miniIMN (miniImagenet) benchmark, we utilized the SSDlite, pre-trained on MS COCO, provided by the PyTorch open library, not the ImageNet pre-trained EffiicientNet. Also, to demonstrate that MuFAN can show excellent online CL performance regardless of the type of pre-trained models, we experimented with various pre-trained models provided by the open libraries and observed consistently excellent performance. The results are summarized in Supplementary F. We hope that this response has been helpful in resolving your concern. If you have any further concerns left, please leave a comment.

---

> > > > ### Comment · Reviewer_aHBY · 2022-11-19
> > > > **Re: Re: Re: Response to Reviewer aHBY (2/2)**
> > > >
> > > > Thank you for the clarification. Using the dataset other than the target dataset (that is used for the evaluation) makes the method fairer to compare with other methods. But it still requires the additional overhead of pretraining. But with that additional overhead, this method might be useful. Thus, I am raising my score to 6.

---

### Decision · Program_Chairs · 2023-01-20

**Decision:**

Accept: poster

**Justification For Why Not Higher Score:**

While the novelty of the contributions are not in question, there are some methodological issues raised by reviewers that bring into question the long-term impact of the work. However, since it is a first-of-its-kind work on exploiting pre-trained encoders for online CIL, such methodological issues can and should be discussed at a forum like ICLR -- as a poster.

**Justification For Why Not Lower Score:**

Again, the novelty of the multiple contributions and the thoroughness of the experimental evaluation earn this work a place at ICLR. Although reviewer opinion was somewhat split, there were no vigorous detractors of the work as submitted.

**Metareview: Summary, Strengths And Weaknesses:**

# Summary of Contribution

This paper describes an approach to online class-incremental learning based on exploiting pre-trained models. The authors argue that a different stability-plasticity tradeoff is needed for online continual learning (i.e. that *stability* is far more important) and propose the MuFAN model which incorporates a new, structure-based distillation to mitigate forgetting and a stability-plasticity normalization layer that tries to simultaneously maintain both stability and plasticity. Experimental results are given on standard online CIL benchmarks and a comparison is made with recent work from the state-of-the-art.

# Strengths

+ **Novel Application of Pre-trained Models**: This paper proposes to fuse features from a pre-trained encoder during online continual learning. While using a pre-trained model seems like an obvious approach to generically improve performance, the authors perform a methodical and well-motivated analysis of its incorporation into an online continual learning pipeline -- and the pre-trained encoder is *not* applied in an obvious way.

+ **Comprehensive Experimental Evaluation**: The reviewers a nearly unanimous in their praise of the experimental evaluation in the submitted manuscript, from the comparison with the state-of-the-art to the comprehensive ablation studies performed.

+ **Novelty**: The paper makes contributions in several, orthogonal directions. Apart from exploiting a pre-trained encoder, the authors also propose a new distillation technique based on the structure of fused, multi-scale features, as well as a new normalization layer specifically designed to maintain stability and plasticity.

# Weaknesses

+ **Clarity and Reproducibility**: Reviewers had a somewhat mixed opinion regarding the clarity of the paper. Some found the paper to be "dense" and "a hard read". In rebuttal the authors worked to improve several aspects regarding clarity.

+ **Computational Burden**: Exploiting pre-training goes somewhat contrary to the philosophy of continual learning -- as pointed out by reviewers -- and entails additional computational costs as well. During the discussion phase the authors provided additional analysis showing that their proposed online CIL pipeline (post pre-training) does incur additional computational costs compared to DualNet (about 10% more), but also results in nearly 10% points in accuracy gain.

# Summary

During the discussion phase there was a lively back-and-forth between the authors and reviewers (and between reviewers and AC). The authors vigorously engaged with reviewers and multiple levels, and provided new experimental results comparing with very recent works (published after the ICLR cutoff). Given the novel contributions of the work (which must be emphasized does *not* simply reduce to "using a pre-trained backbone) in multiple dimensions, the paper merits a place at ICLR as it takes a thoughtful and methodical look at *how* pretrained encoders can and should be exploited for online CIL.


**Note From Pc:**

if the above contains the word "oral" or "spotlight" please see: "oral" presentation means -> notable-top-5% and "spotlight" means -> notable-top-25%. As stated in our emails, we are disassociating presentation type from AC recommendations